# Negative regulation of APC/C activation by MAPK-mediated attenuation of Cdc20[Slp1] under stress

Li Sun[1†], Xuejin Chen[1†], Chunlin Song[1†], Wenjing Shi[1†], Libo Liu[1†], Shuang Bai[1†], Xi Wang[1], Jiali Chen[1], Chengyu Jiang[1], Shuang-min Wang[1], Zhou-qing Luo[1], Ruiwen Wang[2*], Yamei Wang[1*], Quan-wen Jin[1*]

[1]State Key Laboratory of Cellular Stress Biology, School of Life Sciences, Faculty of Medicine and Life Sciences, Xiamen University, Xiamen, China; [2]Institute of Life Sciences, College of Biological Science and Engineering, Fuzhou University, Fuzhou, China

**\*For correspondence:**
ruiwen.wang@fzu.edu.cn (RW);
wangyamei@xmu.edu.cn (YW);
jinquanwen@xmu.edu.cn (Q-wenJ)

[†]These authors contributed equally to this work

**Competing interest:** The authors declare that no competing interests exist.

**Abstract** Mitotic anaphase onset is a key cellular process tightly regulated by multiple kinases. The involvement of mitogen-activated protein kinases (MAPKs) in this process has been established in *Xenopus* egg extracts. However, the detailed regulatory cascade remains elusive, and it is also unknown whether the MAPK-dependent mitotic regulation is evolutionarily conserved in the single-cell eukaryotic organisms such as fission yeast (*Schizosaccharomyces pombe*). Here, we show that two MAPKs in *S. pombe* indeed act in concert to restrain anaphase-promoting complex/cyclosome (APC/C) activity upon activation of the spindle assembly checkpoint (SAC). One MAPK, Pmk1, binds to and phosphorylates Slp1[Cdc20], the co-activator of APC/C. Phosphorylation of Slp1[Cdc20] by Pmk1, but not by Cdk1, promotes its subsequent ubiquitylation and degradation. Intriguingly, Pmk1-mediated phosphorylation event is also required to sustain SAC under environmental stress. Thus, our study establishes a new underlying molecular mechanism of negative regulation of APC/C by MAPK upon stress stimuli, and provides a previously unappreciated framework for regulation of anaphase entry in eukaryotic cells.

## eLife assessment

The regulation of mitosis and the dynamics of the mitotic spindle in it are central to cell division with high fidelity and crucial for normal division and development and defects therein can lead to disease. A key component of ensuring the fidelity is the "spindle assembly checkpoint". This **valuable** study using **convincing** experimental approaches in fission yeast has revealed novel links between the MAP-kinase signalling pathway modulating the spindle assembly checkpoint.

## Introduction

The evolutionarily conserved mitogen-activated protein kinase (MAPK) signaling pathways regulate multiple cellular functions in eukaryotic organisms in response to a wide variety of environmental cues (***Plotnikov et al., 2011***). However, different MAPK pathways have been evolved in an organism to integrate diverse signals and fulfill different regulations on various effectors (***Cansado et al., 2021***; ***Ronkina and Gaestel, 2022***). This makes specifying the MAPKs and substrates involved in a specific physiological process rather complex and challenging.

The fission yeast *Schizosaccharomyces pombe* has three MAPK-signaling cascades: the pheromone signaling pathway (PSP), the stress-activated pathway (SAP), and the cell integrity pathway (CIP),

with Spk1, Sty1, or Pmk1 as the MAPK, respectively (*Figure 1A*; *Cansado et al., 2021*; *Perez and Cansado, 2010*). So far, the only known cell-cycle control stages linked to MAPKs in fission yeast are at $G_2$/M transition and during cytokinesis (*Gómez-Gil et al., 2020*; *Petersen and Hagan, 2005*; *Petersen and Nurse, 2007*). The p38 MAPK family member Sty1 either arrests or promotes mitotic commitment in unperturbed, stressed, or resumed cell cycles after stress depending on the downstream kinases it associates with. Sty1 can phosphorylate kinase Srk1 or Polo kinase Plo1 to negatively or positively regulate Cdc25 phosphatase and thus the onset of mitosis, respectively (*López-Avilés et al., 2005*; *López-Avilés et al., 2008*; *Petersen and Hagan, 2005*; *Petersen and Nurse, 2007*; *Smith et al., 2002*). Recent studies have also demonstrated that Sty1 and Pmk1 can pose negative effect on assembly of contractile actomyosin ring and cytokinesis in response to actin cytoskeleton damage or cell wall stress (e.g. treatment with blankophor, caspofungin, and caffeine) (*Edreira et al., 2020*; *Gómez-Gil et al., 2020*), though the detailed mechanisms have not been clearly elucidated.

In late mitosis, the timely polyubiquitylation and subsequent degradation of securin and cyclin B by anaphase-promoting complex/cyclosome (APC/C) play a critical role for anaphase onset and chromosome segregation (*Peters, 2006*; *Sullivan and Morgan, 2007*; *Yamano, 2019*). Given the essential role of APC/C in triggering chromosome segregation, it is not surprising that APC/C is a key molecular target of the spindle assembly checkpoint (SAC), which is an intricate surveillance mechanism that prolongs mitosis until all chromosomes achieve correct bipolar attachments to spindle microtubules (*McAinsh and Kops, 2023*; *Murray, 2011*; *Musacchio, 2015*). Previous studies have revealed that the active MAPK is required for spindle checkpoint activation and APC/C inhibition in *Xenopus* egg extracts or tadpole cells (*Chen, 2004*; *Chung and Chen, 2003*; *Minshull et al., 1994*; *Takenaka et al., 1997*; *Zhao and Chen, 2006*). These studies suggested that phosphorylation of several components of SAC, including Cdc20, Mps1, and Bub1, by MAPK serves as regulatory signals to block APC/C activation upon SAC activation (*Chen, 2004*; *Chung and Chen, 2003*; *Zhao and Chen, 2006*). However, all these studies used either the anti-MAPK antibodies or a small molecule inhibitor UO126 to inhibit MAPK or MAPK-specific phosphatase MKP-1 to antagonize MAPK activity, these reagents could not distinguish the contributions of different MAPKs with high specificity. Thus, those seemingly defined mechanisms might suffer the ambiguity due to potential specificity issues.

In *S. pombe*, whether MAPKs play key roles in regulation of the APC/C and spindle checkpoint remains largely unexplored. Intriguingly, all genes encoding the major components of three MAPK signaling cascades in fission yeast are non-essential (*Kim et al., 2010*), and this provides the possibility for investigations on their potential functions in a genetic background with deletions of the relevant genes. In this work, we set out to examine directly the requirement of the fission yeast MAPKs in mitotic APC/C activation and anaphase entry. Our results uncovered a previously unappreciated mechanism in which Pmk1, the MAPK of CIP, phosphorylates the APC/C activator Slp1 (the Cdc20 homolog in *S. pombe*) upon SAC activation. Phosphorylation of Slp1$^{Cdc20}$ results in its ubiquitylation and degradation and thus subsequently impedes APC/C activation and anaphase entry. This mechanism also operates in response to spindle defects under environmental stress. Therefore, our finding extends cell-cycle control stages linked to MAPKs from previously recognized $G_2$/M transition and cytokinesis to anaphase entry and mitotic exit in fission yeast. It will be interesting to address whether this MAPK-mediated Cdc20 regulation exists as a general mechanism for balancing spindle checkpoint activity in higher eukaryotes.

## Results
### SAP and CIP signaling components are required for spindle checkpoint activity

As the first step to examine the possible requirement of the fission yeast MAPKs in mitotic progression, we tested the sensitivity of deletion mutants of three fission yeast MAPKs to microtubule-destabilizing drug thiabendazole (TBZ), which has been routinely used as an indicator of defective kinetochore or spindle checkpoint (*Akera et al., 2015*; *Saitoh et al., 1997*). We noticed that *pmk1Δ* and *sty1Δ* but not *spk1Δ* cells were extremely sensitive to TBZ, and constitutively activated CIP or SAP signaling by overexpressing Pek1$^{DD}$ (Pek1-S234D;T238D) (*Sugiura et al., 1999*) or Wis1$^{DD}$ (Wis1-S469D;T473D) (*Shiozaki et al., 1998*), respectively, but not activated PSP signaling by overexpressing Byr1$^{DD}$ (Byr1-S214D;T218D) (*Ozoe et al., 2002*), also caused yeast cells to be sensitive to TBZ (*Figure 1B*;

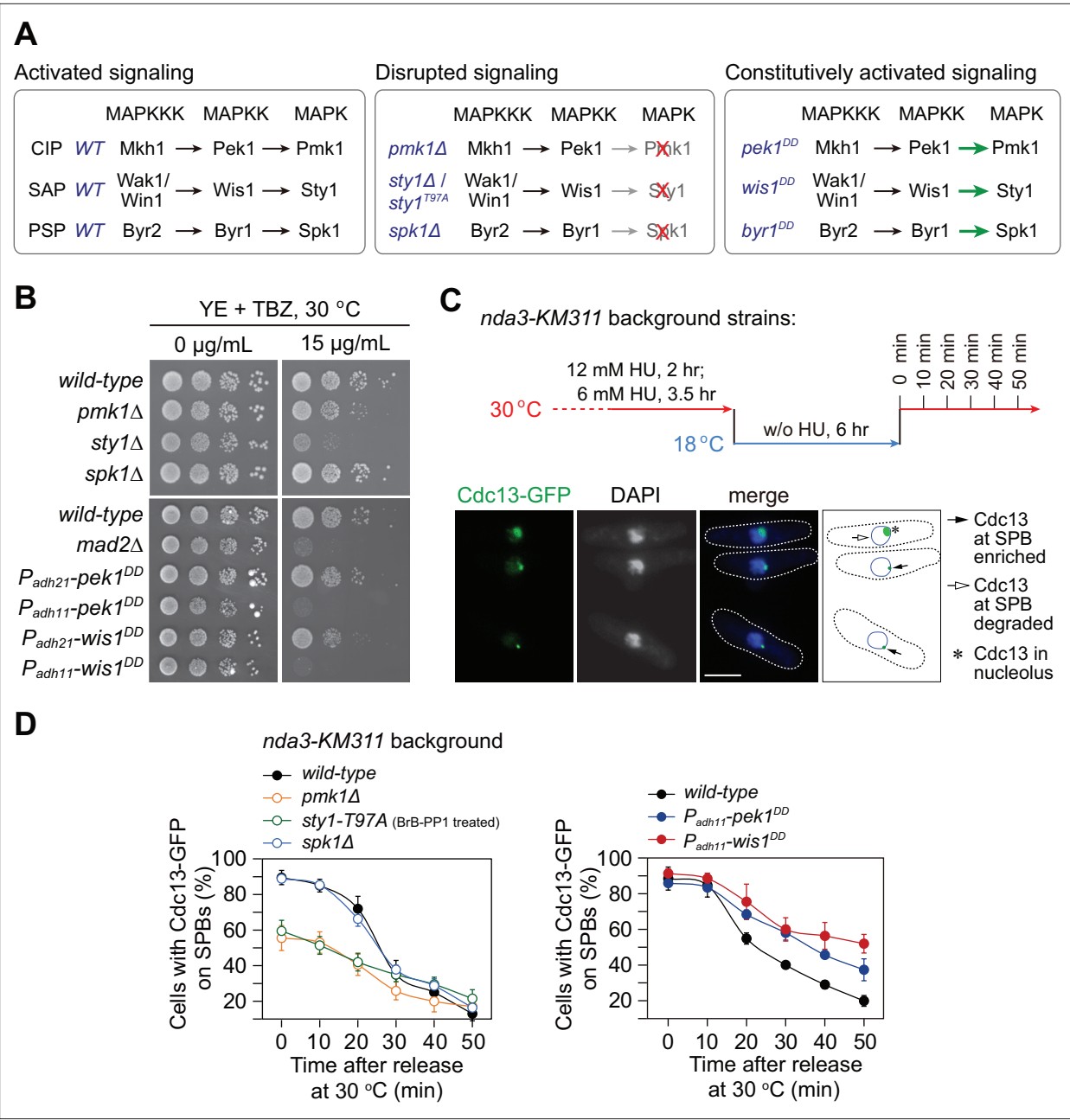

**Figure 1.** *pmk1Δ* and *sty1-T97A* mutants display spindle checkpoint activation defects, while *pek1^DD* and *wis1^DD* mutants are defective in checkpoint silencing. (**A**) Schematic of core modules of three *Schizosaccharomyces pombe* mitogen-activated protein kinase (MAPK) signaling pathways. Each cascade consists of three core kinases (MAP kinase kinase kinase (MAPKKK), MAP kinase kinase (MAPKK), and MAPK). CIP, the cell integrity pathway; SAP, the stress-activated pathway; PSP, pheromone signaling pathway. These pathways can be disrupted by *pmk1Δ*, *sty1Δ*, or (adenosine triphosphate) ATP analog-sensitive mutation *sty1-T97A* or *spk1Δ*, and constitutively activated by mutations in MAPKKs Pek1 (*pek1^DD*, *pek1-S234D;T238D*), Wis1 (*wis1^DD*, *wis1-S469D;T473D*), or Byr1 (*byr1^DD*, *byr1-S214D;T218D*), respectively. (**B**) Serial dilution assay on thiabendazole (TBZ) sensitivity of all MAPK deletion mutants and *pek1^DD*- or *wis1^DD*-overexpressing mutants. *mad2Δ* is a positive control. Note *P_adh11* is a stronger version of *P_adh21* promoter. (**C**) Schematic depiction of the experiment design for time-course analyses on spindle assembly checkpoint (SAC) or anaphase-promoting complex/cyclosome (APC/C) activation. *nda3-KM311* cells carrying Cdc13-GFP were grown, synchronized with hydroxyurea (HU) and treated at 18°C for 6 hr to activate SAC, and finally shifted back to the permissive temperature 30°C. Samples were collected at 10 min intervals and subjected to microscopy analyses. Example pictures of cells with Cdc13-GFP signals enriched or disappeared at spindle pole bodies (SPBs) are shown. Scale bar, 5 μm. (**D**) Time-course analyses of SAC activation and inactivation in *nda3-KM311 cdc13-GFP* strains with indicated genotypes. *sty1-T97A* was inactivated by 5 μM 3-BrB-PP1. For each time point, ≥300 cells were counted for every sample. The experiment was repeated ≥3 times and error bars indicate mean ± standard deviation.

The online version of this article includes the following source data and figure supplement(s) for figure 1:

*Figure 1 continued on next page*

*Figure 1 continued*

**Source data 1.** Raw data of time-course analyses of Cdc13-GFP at spindle pole body (SPB) for *Figure 1D*.

**Figure supplement 1.** Examination of protein levels of overexpressed Pek1$^{DD}$ and Wis1$^{DD}$ and analyses of spindle checkpoint inactivation efficiency in $P_{adh21}$-*pek1*$^{DD}$ and $P_{adh21}$-*wis1*$^{DD}$ mutants.

**Figure supplement 1—source data 1.** Uncropped blots for *Figure 1—figure supplement 1A*.

**Figure supplement 1—source data 2.** Raw data of time-course analyses of Cdc13-GFP at spindle pole body (SPB) for *Figure 1—figure supplement 1B, C*.

**Figure supplement 1—source data 3.** Full raw unedited blot (anti-HA) for *Figure 1—figure supplement 1A*.

**Figure supplement 1—source data 4.** Full raw unedited blot (Cdc2) for *Figure 1—figure supplement 1A*.

**Figure supplement 2.** Characterization of possible effect of overexpressed Byr1$^{DD}$ on spindle checkpoint activation and inactivation.

**Figure supplement 2—source data 1.** Uncropped blots for *Figure 1—figure supplement 2A*.

**Figure supplement 2—source data 2.** Raw data of time-course analyses of Cdc13-GFP at spindle pole body (SPB) for *Figure 1—figure supplement 2C, D*.

**Figure supplement 2—source data 3.** Full raw unedited blot (anti-HA) for *Figure 1—figure supplement 2A*.

**Figure supplement 2—source data 4.** Full raw unedited blot (Cdc2) for *Figure 1—figure supplement 2A*.

**Figure supplement 3.** Analyses of metaphase arrest efficiency in *pmk1Δ*, *sty1-T97A*, and *spk1Δ* mutants upon Mad2 overexpression.

**Figure supplement 3—source data 1.** Raw data of quantitative analyses of cells with short spindles in response to Mad2 overexpression for *Figure 1—figure supplement 3B*.

*Figure 1—figure supplement 1A* and *Figure 1—figure supplement 2A, B*). These observations indicated that Pmk1 and Sty1 pathways might be involved in spindle checkpoint-related processes.

To extend our analysis and precisely quantify whether the SAC is properly activated and maintained in MAPK mutants, we adopted one well-established assay using the cold-sensitive β-tubulin mutant *nda3-KM311*, which compromises the kinetochore–spindle microtubule attachment upon cold treatment (*Hiraoka et al., 1984*), to analyze the ability of cells to arrest in mitosis in the absence of spindle microtubules and to enter anaphase when spindle microtubules are reassembled (*Bai et al., 2022*; *May et al., 2017*; *Vanoosthuyse and Hardwick, 2009*). In this assay, the SAC was first robustly activated by the *nda3-KM311* mutant at 18°C and then inactivated simply by shifting mitotically arrested cells back to permissive temperature (30°C) (*Figure 1C*). Because Cdc13 (cyclin B in *S. pombe*) localizes to the spindle pole bodies (SPBs) in early mitosis and is degraded by APC/C to promote metaphase–anaphase transition, accumulation of Cdc13-GFP on SPBs after cold treatment serves as the read-out of the SAC activation, and the disappearance rate of Cdc13-GFP spot under permissive temperature reflects the SAC inactivation efficacy (*Figure 1C*). It is noteworthy that the ATP analog-sensitive mutant *sty1-T97A* (i.e. *sty1-as2*) (*Zuin et al., 2010*), instead of *sty1Δ* deletion mutant, was used in this assay. That was due to the G$_2$ delay in the *sty1Δ* deletion mutant (*Shiozaki and Russell, 1995a*), which could interfere with mitotic timing for further analysis in this assay. We found that inactivation of CIP signaling by *pmk1Δ* or SAP by *sty1-T97A* compromised full SAC activation, as these two mutants had only less than 60% of cells with Cdc13-GFP on SPBs after cold treatment for 6 hr, whereas the percentage in wild-type and *spk1Δ* cells after the same treatment was roughly 90% (*Figure 1D*; *Figure 1—figure supplement 1C*). Consistently, the disappearance of Cdc13-GFP from SPBs when released from metaphase arrest was severely delayed in *pek1*$^{DD}$ and *wis1*$^{DD}$ but not *byr1*$^{DD}$ mutant cells (*Figure 1D*; *Figure 1—figure supplement 1B* and *Figure 1—figure supplement 2C, D*), suggesting delayed SAC inactivation and anaphase onset after release from spindle checkpoint arrest when CIP or SAP signaling is constitutively activated.

To corroborate our above observations of negative effect of CIP or SAP signaling on SAC strength, we took advantage of a different spindle checkpoint activation and metaphase-arresting mechanism involving Mad2 overexpression (*May et al., 2017*). After $P_{nmt1}$-*mad2*$^+$ was induced to overexpress for 18 hr, we counted cells with short spindles visualized by GFP-Atb2 as indicative of metaphase arrest and SAC activation (*Figure 1—figure supplement 3A*). We found that the ability of Mad2-overexpressing cells to arrest in mitosis was clearly compromised in *pmk1Δ* and *sty1-T97A* mutants, but not in *spk1Δ* mutant (*Figure 1—figure supplement 3B*).

All above data suggested that MAPK pathways are evolutionarily conserved in the fission yeast to participate in spindle checkpoint activation and maintenance, in which the CIP and SAP signaling pathways, but not the Spk1 pathway, are actively involved.

## CIP and SAP signalings attenuate protein levels of Slp1$^{Cdc20}$ and facilitate association of mitotic checkpoint complex with APC/C, respectively

The mitotic checkpoint complex (MCC, consisting of Mad3-Mad2-Slp1$^{Cdc20}$ in *S. pombe*) is the most potent inhibitor of the APC/C prior to anaphase (*Kapanidou et al., 2017*; *Primorac and Musacchio, 2013*; *Sudakin et al., 2001*). Previous study has revealed that *Xenopus* MAPK phosphorylates Cdc20 and facilitates its binding by Mad2, BubR1 (Mad3 in *S. pombe*), and Bub3 to form a tight MCC to prevent Cdc20 from activating the APC/C (*Chung and Chen, 2003*). In fission yeast, the key SAC components Mad2 and Mad3 and one molecule of Slp1$^{Cdc20}$ form MCC which binds to APC/C through another molecule of Slp1$^{Cdc20}$ upon checkpoint arrest, and the recovery from mitotic arrest accompanies the loss of MCC–APC/C binding (*May et al., 2017*; *Sczaniecka et al., 2008*; *Sewart and Hauf, 2017*; *Vanoosthuyse and Hardwick, 2009*).

Since we found that constitutive activation of CIP or SAP signaling in fission yeast by *pek1$^{DD}$* or *wis1$^{DD}$* mutations caused prolonged SAC activation, we speculated that these two MAPK pathways might also execute their functions cooperatively through a similar mechanism to that in *Xenopus*. To test this possibility, we analyzed the MCC occupancy on APC/C by immunoprecipitations of the APC/C subunit Lid1 (*Yoon et al., 2002*) in *pek1$^{DD}$* and *wis1$^{DD}$* cells arrested by activated checkpoint. Compared to wild-type, more MCC components (Mad2 plus Mad3 and Slp1$^{Cdc20}$) were co-immunoprecipitated in *wis1$^{DD}$* cells, while less of them were co-immunoprecipitated in *pek1$^{DD}$* cells (*Figure 2A*), suggesting that activation of the SAP but not the CIP pathway enhanced the association of MCC with APC/C.

In the course of the above immunoprecipitations, we noticed the decrease of co-immunoprecipitated Mad2 and Mad3 in *pek1$^{DD}$* cells, which was largely out of expectation. Further analysis of the inputs of these immunoprecipitations showed that Slp1$^{Cdc20}$ levels were significantly reduced in *pek1$^{DD}$* but not *wis1$^{DD}$* cells (*Figure 2A*). Our immunoblotting analysis using the total lysate of *pek1$^{DD}$* or *wis1$^{DD}$* cells arrested at metaphase also confirmed these results (*Figure 2B*). Consistently, the Slp1$^{Cdc20}$ level was significantly increased in *pmk1Δ* and *pmk1Δ sty1-T79A* cells but not in *sty1-T79A* cells (*Figure 2C*). The elevated Slp1$^{Cdc20}$ level in *pmk1Δ* cells is similar to that we observed in cells harboring two copies of *slp1$^{+}$*, which leads to artificially increased Slp1$^{Cdc20}$ level and also causes an SAC strength defect (*Bai et al., 2022*). It is noteworthy that attenuated levels of Slp1$^{Cdc20}$ in *pek1$^{DD}$* cells was not due to decreased transcription of *slp1$^{+}$* gene (*Figure 2D*). These results collectively revealed a negative effect of CIP but not SAP signaling on Slp1$^{Cdc20}$ levels upon SAC activation, which potentially results in the weakened MCC–APC/C interaction (*Figure 2A*).

Next, we also tested whether activation of the CIP signaling causes downregulation of Slp1$^{Cdc20}$ in normally growing mitotic cells, which were enriched by release of *cdc25-22* background cells from G$_2$/M-arrest (*Figure 2—figure supplement 1A, B*). Intriguingly, we could only detect marginally reduced Slp1$^{Cdc20}$ levels in *pek1$^{DD}$* cells, which was consistent with the slightly retarded anaphase entry in those cells, but we failed to observe any elevated Slp1$^{Cdc20}$ levels in *pmk1Δ* cells (*Figure 2—figure supplement 1C*). These results suggested that CIP signaling only plays a negligible role in influencing Slp1$^{Cdc20}$ levels when SAC is not activated.

Together, our above data suggested a new mechanism involves the division of labor between the two fission yeast MAPK pathways CIP (Pek1-Pmk1) and SAP (Wis1-Sty1) upon SAC activation, which is required in concert to lower Slp1$^{Cdc20}$ levels and enhance MCC affinity for APC/C to strongly inhibit APC/C activity (*Figure 2E*). In this study, we only focused on characterizing the role of the CIP/Pmk1 in regulating SAC inactivation and anaphase onset hereafter.

## Pmk1 directly binds Slp1$^{Cdc20}$ to attenuate its levels upon SAC activation

To gain further insight into how Slp1$^{Cdc20}$ levels are modulated by Pmk1, we examined the potential interaction of Pmk1 with Slp1$^{Cdc20}$. Through incubating MBP-tagged Slp1$^{Cdc20}$ with metaphase-arrested

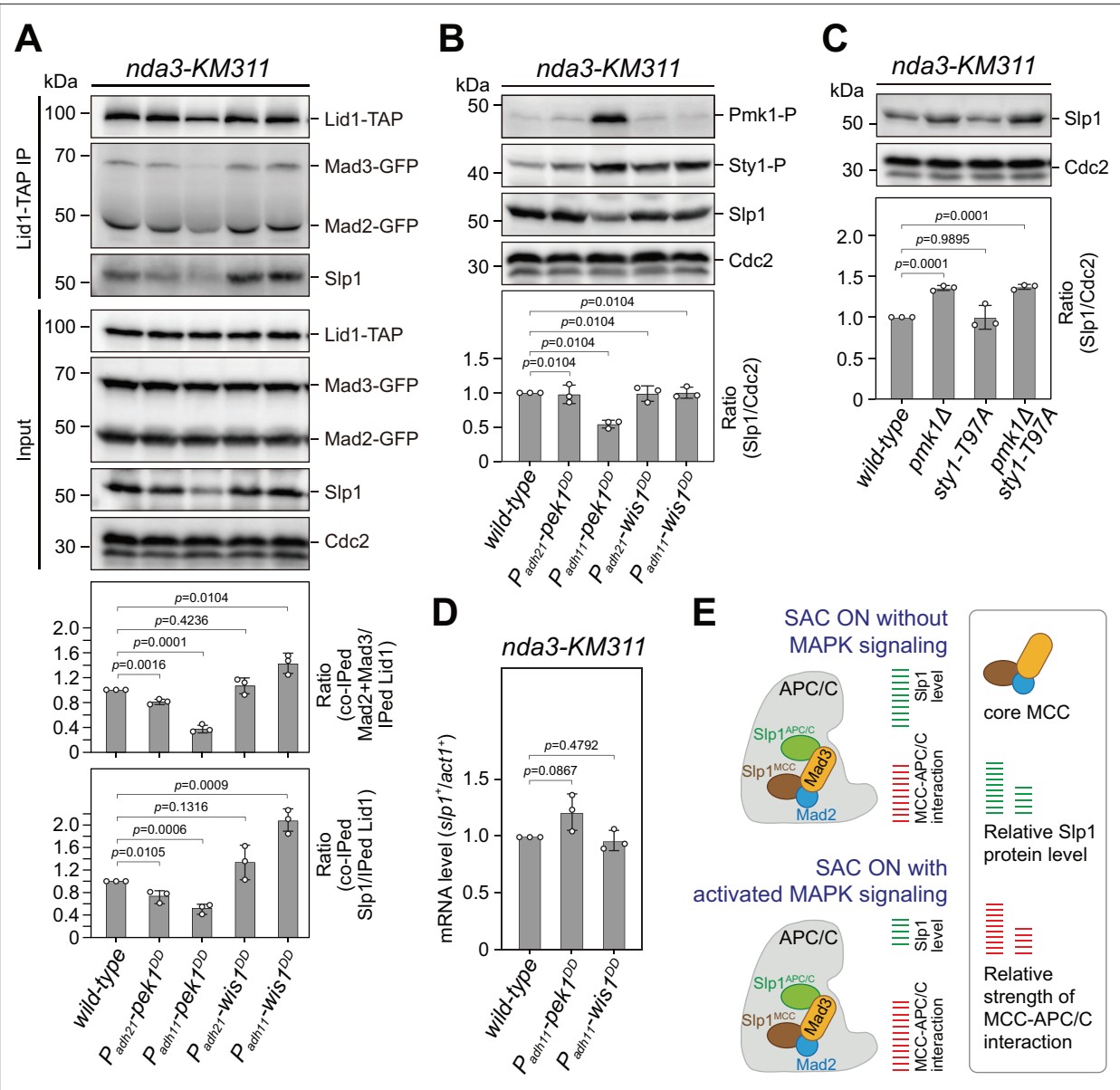

**Figure 2.** Upon spindle checkpoint activation, Slp1[Cdc20] levels are reduced in *pek1[DD]* and the mitotic checkpoint complex (MCC)–anaphase-promoting complex/cyclosome (APC/C) association is enhanced in *wis1[DD]* cells compared to wild-type cells. (**A**) Co-immunoprecipitation analysis on MCC–APC/C association. Cells with indicated genotypes were grown at 30°C to mid-log phase and arrested at 18°C for 6 hr. Lid1-TAP was immunoprecipitated and associated Mad2, Mad3, and Slp1[Cdc20] were detected by immunoblotting. The amount of co-immunoprecipitated Mad2, Mad3, and Slp1[Cdc20] was quantified by being normalized to those of total immunoprecipitated Lid1 in each sample, with the relative ratio between Mad2-GFP plus Mad3-GFP or Slp1[Cdc20] and Lid1-TAP in wild-type sample set as 1.0. Blots are representative of three independent experiments. p values were calculated against wild-type cells. (**B, C**) Immunoblot analysis of Slp1[Cdc20] abundance in *nda3-KM311* cells treated at 18°C for 6 hr. Slp1[Cdc20] levels were quantified with the relative ratio between Slp1[Cdc20] and Cdc2 in wild-type strain set as 1.0. Phosphorylated Pmk1 (Pmk1-P) or phosphorylated Sty1 (Sty1-P) in (**A**) were detected using anti-phospho p42/44 and anti-phospho p38 antibodies and represents activated cell integrity pathway (CIP) or stress-activated pathway (SAP) signaling, respectively. *sty1-T97A* was inactivated by 5 μM 3-BrB-PP1. Blots shown are the representative of three independent experiments. p values were calculated against wild-type cells. (**D**) Real time quantitative PCR (RT-qPCR) analysis of mRNA levels of *slp1+*. Cells with indicated genotypes were grown and treated as in (**A–C**) before RNA extraction. The relative fold-change (*slp1+*/*act1+*) in mRNA expression was calculated with that in wild-type cells being normalized to 1.0. Note mRNA level of *slp1+* in *pek1[DD]* mutant is not decreased. Error bars indicate mean ± standard deviation of three independent experiments. Two-tailed unpaired *t*-test was used to derive p values. (**E**) Schematic summary of the negative effect of activated CIP and SAP signaling on APC/C activation based on primary phenotype characterization of *pmk1Δ*, *sty1-T97A*, *pek1[DD]*, and *wis1[DD]* mutants.

The online version of this article includes the following source data and figure supplement(s) for figure 2:

**Source data 1.** Uncropped blots for *Figure 2A–C*.

*Figure 2 continued on next page*

*Figure 2 continued*

**Source data 2.** Raw data of co-immunoprecipitation rate of Mad2/Mad3/Slp1, Slp1 level measurement, and RT-qPCR for *Figure 2A–D*.

**Source data 3.** Raw unedited blots for *Figure 2*.

**Figure supplement 1.** Examination of protein levels of Slp1^Cdc20 in normally growing mitotic *pmk1Δ* or*pek1DD* cells.

**Figure supplement 1—source data 1.** Uncropped blots for *Figure 2—figure supplement 1C*.

**Figure supplement 1—source data 2.** Raw data of time-course analyses of arrest-and-release of *cdc25-22* mutants for *Figure 2—figure supplement 1B*.

**Figure supplement 1—source data 3.** Full raw unedited blot (Slp1) for *Figure 2—figure supplement 1C*.

**Figure supplement 1—source data 4.** Full raw unedited blot (Cdc2) for *Figure 2—figure supplement 1C*.

yeast cell lysates, we found that Pmk1 expressed from yeast cells could be pull-downed by Slp1^Cdc20 (*Figure 3A*).

In previous reports, MAPK-docking sites, which are important for interaction with MAPKs, have been identified at the N termini of different MAPK kinases (MAPKKs) from yeast to humans with a notable feature of a cluster of at least two basic residues (mainly Lys/K and Arg/R) and a hydrophobic-X-hydrophobic sequence separated by a spacer of two to six residues (*Bardwell et al., 2001*; *Bardwell and Thorner, 1996*). Thus, we attempted to identify the potential key residues in Slp1^Cdc20 that mediate the Pmk1–Slp1^Cdc20 interaction. By visual scanning of Slp1^Cdc20 sequence, we noticed five basic-residue patches which are also present in several fungal species and loosely resemble the consensus sequence of the known MAPK-docking sites, and four of them are within the N-terminal portion of the protein (*Figure 3B* and *Figure 3—figure supplement 1*). We found that mutating either of the two most N-terminal basic-residue patches (19-KKR-21 and 47-KR-48) to glutamates (Glu, E) or alanines (Ala, A) in Slp1^Cdc20 resulted in decreased amount of Slp1^Cdc20 pulled down by Pmk1-GST in vitro using the bacterially expressed recombinant proteins, and mutating both patches or mutating only the two arginine residues (R21 and R48) in those patches was sufficient to completely disrupt their interaction (*Figure 3C* and *Figure 3—figure supplement 2*). Importantly, mutations of these Pmk1-docking sequences in Slp1 (PDSS for short) to alanines mimicked *pmk1Δ* mutant in both elevated Slp1^Cdc20 protein levels (*Figure 3D*) and defective activation and maintenance of the SAC upon *nda3*-mediated checkpoint arrest (*Figure 3E* and *Figure 3—figure supplement 3*). Also, similar to removal of Pek1 downstream MAPK Pmk1, the lowered Slp1^Cdc20 protein levels and delayed SAC inactivation in *pek1DD* mutant could be relieved by *slp1(DS-5A)* mutant, which is deficient in Pmk1-docking (*Figure 3D, E* and *Figure 3—figure supplement 3*).

Previously, one conserved non-canonical Pmk1-docking sequence which bears positively charged residues and IYT motif has been identified in MAPK phosphatase Pmp1 (*Sacristán-Reviriego et al., 2014*). Very strikingly, insertion of a 15 amino acid sequence of Pmp1 containing the Pmk1-docking sequence (or PDSP for short) right in front of the first basic-residue patch in Slp1(DS-5A) mutant protein restored its interaction with Pmk1 (*Figure 3B, C*). And this artificially forced Pmk1–Slp1^Cdc20 interaction via PDSP was able to lower Slp1^Cdc20 protein levels and restore higher percentage of metaphase arrest upon SAC activation in *slp1(DS-5A)* mutant, which is Pmk1-docking-deficient, though the rescuing effect was less efficient than that in the presence of genuine PDSS (*Figure 3D, E* and *Figure 3—figure supplement 3*). These data firmly supported the idea that Pmk1 binds Slp1^Cdc20 to downregulate its levels upon SAC activation.

Collectively, these results demonstrated that Pmk1 directly binds to Slp1^Cdc20 through the two short N-terminal basic patches, which serve as the major docking sites for Pmk1, and the physical interaction of Pmk1–Slp1^Cdc20 is involved in attenuation of Slp1^Cdc20 abundance upon SAC activation (*Figure 3F*).

## Pmk1 synergizes with Cdk1 to phosphorylate Slp1^Cdc20

The association of Pmk1 with Slp1^Cdc20 and the negative effect of constitutively activated CIP signaling on SAC inactivation relies on Pmk1 kinase suggested that Slp1^Cdc20 could be a direct substrate of Pmk1. To test this possibility, we performed in vitro phosphorylation reactions and found that bacterially expressed recombinant GST-Slp1^Cdc20 could be efficiently phosphorylated in the presence of Pmk1-HA-His purified from yeast cells (*Figure 4A*).

To identify potential phosphorylation sites in vivo, we purified Apc15-GFP from wild-type, *pmk1Δ* or *pek1DD* cells arrested by activated checkpoint with intention that the Slp1^Cdc20-containing APC/C

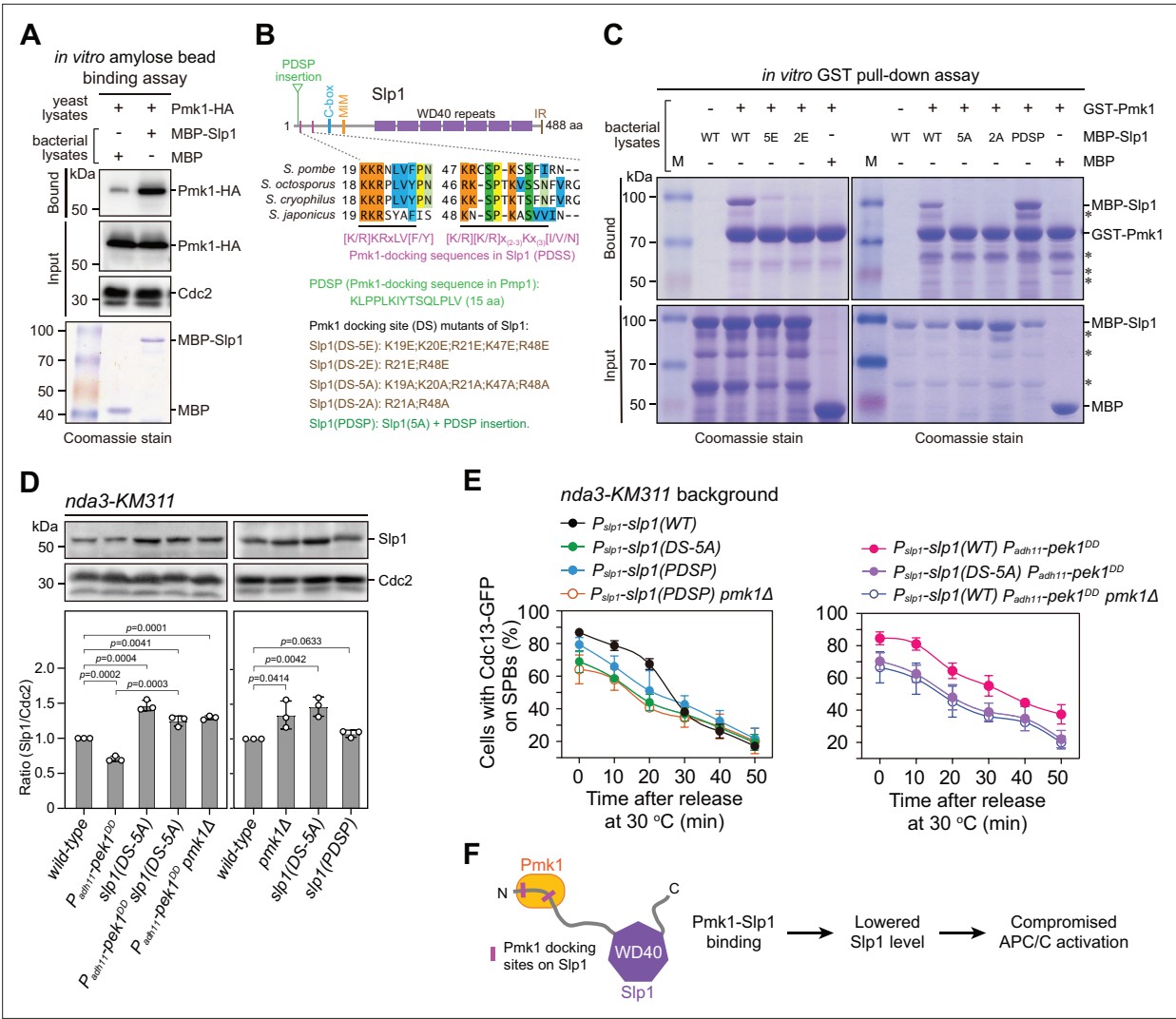

**Figure 3.** Pmk1 directly binds Slp1[Cdc20] to attenuate its levels upon spindle checkpoint activation. (**A**) In vitro binding assay using bacterially expressed MBP-Slp1[Cdc20] and yeast lysates prepared from *nda3-KM311 pmk1-HA-6His* cells arrested at 18°C for 6 hr. Note that weak band detected in MBP sample was due to unspecific background binding to amylose beads. Coomassie blue staining shows inputs for MBP and MBP-Slp1[Cdc20]. (**B**) Schematic depiction of the *S. pombe* Slp1[Cdc20] protein structure with the positions of two confirmed basic-residue patches mediating Slp1[Cdc20]–Pmk1 association indicated by pink bars. Alignment highlights the conservation of basic-residue patches within four *Schizosaccharomyces* species. The deduced Pmk1-docking motifs and different versions of motif mutations are shown. MIM, Mad2-interaction motif; IR, isoleucine–arginine tail. (**C**) In vitro GST pull-down assays with bacterially expressed recombinant GST-Pmk1 and MBP fusions of wild-type Slp1[Cdc20] or Slp1[Cdc20] mutants harboring Pmk1-docking motif mutations. An aliquot of the same amount of MBP-Slp1[Cdc20] as that added in each GST pull-down reaction was immobilized by amylose resin as the input control. Asterisks indicate unspecific or degraded protein bands. (**D**) Immunoblot analysis of Slp1[Cdc20] abundance in *nda3-KM311* cells with indicated genotypes treated at 18°C for 6 hr. Slp1[Cdc20] levels were quantified as in ***Figure 2B, C***. The experiment was repeated three times. The mean value for each sample was calculated, and p values were calculated against wild-type or *pek1[DD]* cells. (**E**) Time-course analyses of spindle assembly checkpoint (SAC) activation and inactivation in *nda3-KM311 cdc13-GFP* strains with indicated genotypes. For each time point, ≥300 cells were counted for every sample. The experiment was repeated three times and the mean value for each sample was calculated as in ***Figure 1D***. (**F**) Schematic summarizing the negative effect of Pmk1–Slp1[Cdc20] association on Slp1[Cdc20] abundance and anaphase-promoting complex/cyclosome (APC/C) activation.

The online version of this article includes the following source data and figure supplement(s) for figure 3:

**Source data 1.** Uncropped blots for ***Figure 3A, C, D***.

**Source data 2.** Raw data of Slp1 level measurement and time-course analyses of Cdc13-GFP at spindle pole body (SPB) for ***Figure 3D, E***.

**Source data 3.** Raw unedited blots for ***Figure 3***.

**Figure supplement 1.** Conservation of basic-residue patches and phosphorylation or ubiquitylation sites within Slp1 homologs in fungi species.

**Figure supplement 2.** Screen for basic-residue patches in Slp1 mediating its direct interaction with Pmk1 by in vitro binding assay.

*Figure 3 continued on next page*

*Figure 3 continued*

**Figure supplement 2—source data 1.** Uncropped gels for *Figure 3—figure supplement 2B*.

**Figure supplement 2—source data 2.** Full raw unedited Coomassie gel (bead-bound GST-Pmk1 and MBP-Slp1) for *Figure 3—figure supplement 2B*.

**Figure supplement 2—source data 3.** Full raw unedited Coomassie gel (MBP-Slp1 input) for *Figure 3—figure supplement 2B*.

**Figure supplement 3.** Quantification of spindle checkpoint inactivation rate in *slp1* mutants with Pmk1-docking site mutations at 0 and 50 min after release at 30°C from *nda3*-mediated arrest.

**Figure supplement 3—source data 1.** Raw data of time-course analyses of Cdc13-GFP at spindle pole body (SPB) for *Figure 3—figure supplement 3*.

complexes could be captured (*Figure 4—figure supplement 1A, B*). Purifications were performed in the presence of phosphatase inhibitors to enrich phosphopeptides. Mass spectrometry analysis revealed that the phosphorylation of Thr480 residue at the most C-terminus of Slp1$^{Cdc20}$ was dependent on the presence of Pmk1, and this modification was present in *pek1$^{DD}$* but not in wild-type cells (*Figure 4B*, *Figure 4—figure supplement 1C–E* and *Figure 4—figure supplement 2*), indicating that phosphorylation of Thr480 is very likely triggered by CIP signaling. To prove this, we raised phospho-specific antibodies against pT480, they could recognize Slp1$^{Cdc20}$ fragment phosphorylated by Pmk1 in vitro, and mutation of Thr480 to alanine (T480A) abolished site-specific phosphorylation (*Figure 4C*). By using these antibodies, we also confirmed that endogenous Slp1$^{Cdc20}$ was indeed phosphorylated at Thr480 in metaphase-arrested *mts3-1* cells, which was enhanced in *pek1$^{DD}$* mutant and diminished in *pmk1Δ* mutant (*Figure 4D*). Furthermore, we examined how in vivo Slp1$^{Cdc20}$ phosphorylation at Thr480 is affected by forced targeting of Slp1$^{Cdc20}$ to Pmk1. By taking advantage of the high binding affinity of green fluorescent protein (GFP)-binding protein (GBP) to GFP and its variants (*Chen et al., 2017*; *Rothbauer et al., 2006*), we constructed strains ectopically expressing Slp1$^{(1–60aa)}$-mEGFP-2xNLS-GST-Slp1$^{(456–488aa)}$ and Pmk1-GBP-mCherry, in which Slp1$^{Cdc20}$ fragments would be artificially tethered to Pmk1 inside the nuclei (*Figure 4—figure supplement 3A, B*). Immunoblot detection of Slp1$^{Cdc20}$ phosphorylation at T480 in vivo demonstrated that indeed the presence of Pmk1-GBP-mCherry could increase pThr480 levels of mEGFP fusion of Slp1$^{Cdc20}$ fragments, and simultaneously overexpressing Pek1$^{DD}$ from *P$_{adh11}$-6xHA-pek1$^{DD}$* further enhanced the effect (*Figure 4—figure supplement 3C*). Thus, our results established Thr480 of Slp1$^{Cdc20}$ as a major Pmk1 phosphorylation site in vivo upon SAC activation.

In addition to pT480, our mass spectrometry analysis also identified Ser28, Thr31, and Ser59 as Pmk1-independent and Ser76 as Pmk1-dependent phosphorylation sites (*Figure 4B*, *Figure 4—figure supplement 1C–E*, and *Figure 4—figure supplement 2*). All these four residues are located within the N-terminal unstructured region, fitting with the Cdk1 core motif (SP or TP), and they are conserved only in four *Schizosaccharomyces* species and several other fungal species but not in higher eukaryotes (*Figure 4B*, *Figure 3—figure supplement 1B*, and *Figure 4—figure supplement 4*). Interestingly, they were also proposed as Cdk1 phosphorylation sites based on quantitative phosphoproteomic analysis in a recent study (*Swaffer et al., 2018*; *Figure 4B*). By using the phospho-specific antibodies against both pS28 and pT31, we could detect the phosphorylation of Ser28 and Thr31 in vitro by Cdc13 (cyclin B)-containing Cdk1 complexes purified from metaphase-arrested yeast cells, which was severely diminished when the kinase activity of ATP analog-sensitive Cdk1 [Cdc2-as1(F84G)] was inhibited by 1-NM-PP1 or when Slp1$^{(S28A;T31A)}$ was added as the substrate in in vitro kinase reactions (*Figure 4E*). Next, we examined whether Ser28 and Thr31 were phosphorylated in vivo. Slp1$^{Cdc20}$ was co-immunoprecipitated with Apc15 from *nda3*-arrested yeast cells and detected with phospho-specific antibodies. We observed phosphorylation of Ser28/Thr31 in wild-type cells, but it disappeared in phosphatase-treated wild-type samples and in 1-NM-PP1-treated *cdc2-as1* cells, or when cells expressing Slp1$^{(S28A;T31A)}$ were collected as the substrate (*Figure 4F*). Thus, Slp1$^{Cdc20}$ is also phosphorylated by Cdk1 at least at Ser28 and Thr31 in mitotic cells of the fission yeast.

We wondered whether Slp1$^{Cdc20}$ phosphorylation by Pmk1 affects its stability and APC/C activation. Indeed, phospho-deficient Slp1$^{T480A}$ mutant showed elevated Slp1$^{Cdc20}$ levels in both metaphase-arrested and asynchronously grown cells (*Figure 4G* and *Figure 4—figure supplement 5*) and compromised SAC activation compared to wild-type cells, likely due to more efficient APC/C activation (*Figure 4H* and *Figure 4—figure supplement 6*). Furthermore, Slp1$^{T480A}$ mutant also reversed the inhibitory effect of *pek1$^{DD}$* on Slp1$^{Cdc20}$ stability and APC/C activation after release from SAC-mediated arrest (*Figure 4G, H*).

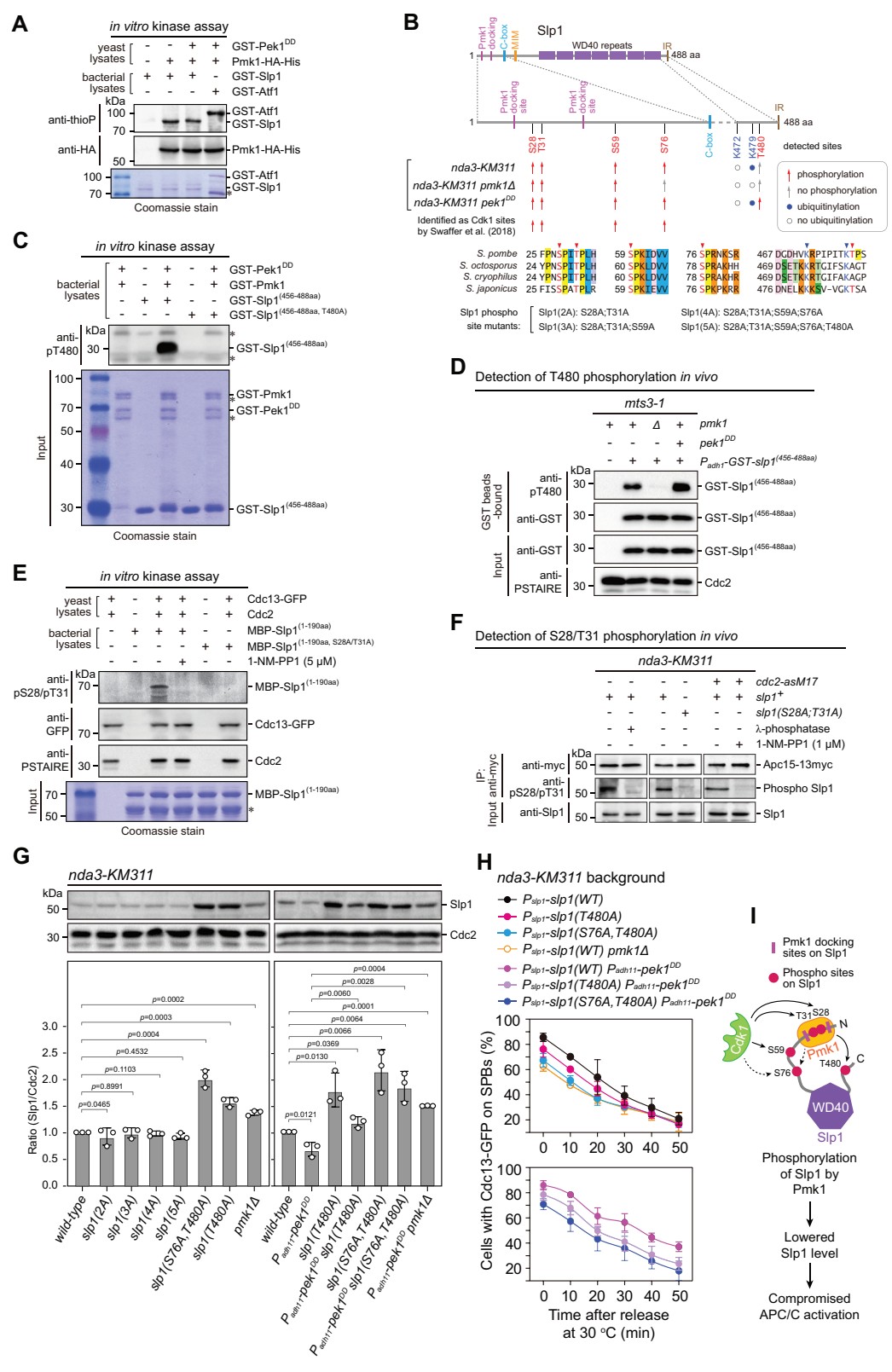

**Figure 4.** Pmk1 synergizes with Cdk1 to phosphorylate and reduce Slp1[Cdc20] abundance. (**A**) Non-radioactive in vitro phosphorylation assays with bacterially expressed recombinant GST-Slp1[Cdc20] and Pmk1-HA-His purified from yeast cells. The incorporation of the thiophosphate group was determined using anti-thiophosphate ester antibodies (anti-thioP) as indicative of phosphorylation. Note that the presence or absence of GST-Pek1[DD] does

*Figure 4 continued on next page*

*Figure 4 continued*

not affect Slp1$^{Cdc20}$ phosphorylation efficiency. A known Pmk1 substrate Atf1 was used as a positive control (Atf1-P). Asterisks indicate bands corresponding to unspecific or likely degraded proteins. (**B**) Summary of mass spectrometry data on Slp1$^{Cdc20}$ phosphorylation and ubiquitylation in vivo in *nda3-KM311*-arrested cells. Red arrows and filled blue circles denote all detected phosphorylated or ubiquitylated sites, respectively, while gray arrows and unfilled circles indicate the absence of phosphorylation or ubiquitylation in some of these sites, respectively. Alignment highlights the conservation of most of the detected phosphorylation and ubiquitylation sites within four *Schizosaccharomyces* species. (**C**) In vitro phosphorylation assays with bacterially expressed recombinant GST-fusion of Slp1$^{Cdc20}$ fragment (456–488aa), GST-Pmk1 and GST-Pek1$^{DD}$. The reactions were blotted with pT480 antibodies. Asterisks indicate bands corresponding to unspecific or likely degraded proteins. (**D**) Immunoblot detection of Slp1$^{Cdc20}$ phosphorylation at T480 in vivo. GST-slp1(456–488aa) was purified from *mts3-1* cells with indicated genotypes arrested at 36°C for 3.5 hr, and detected with anti-GST and anti-pThr480 antibodies. Note that pThr480 is absent in *pmk1Δ* cells, and enhanced in *pek1$^{DD}$* cells relative to that in wild-type cells. (**E**) In vitro phosphorylation assays with bacterially expressed recombinant MBP-fusion of Slp1$^{Cdc20}$ fragment (1–190aa) and Cdc13 (cyclin B)-containing Cdk1 complexes purified from metaphase-arrested *nda3-KM311* yeast cells. 1-NM-PP1 was added as inhibitor for analog-sensitive Cdc2-as. The reactions were blotted with pS28/pT31 antibodies. Asterisks indicate bands corresponding to unspecific or likely degraded proteins. (**F**) Immunoblot detection of Slp1$^{Cdc20}$ phosphorylation at S28/T31 in vivo. Apc15-13myc was immunoprecipitated from *nda3-KM311* cells treated at 18°C for 6 hr and the samples were blotted with pS28/pT31 antibodies. One IP sample from wild-type background was treated with $\lambda$-phosphatase. For *cdc2-asM17* cells, 1-NM-PP1 was added to inactivate Cdc2 during culturing. (**G**) Immunoblot analysis of Slp1$^{Cdc20}$ abundance in *nda3-KM311* cells treated at 18°C for 6 hr. Slp1$^{Cdc20}$ levels were quantified as in *Figure 2B, C*. The experiment was repeated three times. The mean value for each sample was calculated, and p values were calculated against wild-type or *pek1$^{DD}$* cells. (**H**) Time-course analyses of spindle assembly checkpoint (SAC) activation and inactivation in *nda3-KM311 cdc13-GFP* strains with indicated genotypes. For each time point, ≥300 cells were counted for every sample. The experiment was repeated three times and the mean value and p value for each sample were calculated as in *Figure 1D*. (**I**) Schematic summarizing the negative effect of Slp1$^{Cdc20}$ phosphorylation by Pmk1 on its abundance and anaphase-promoting complex/cyclosome (APC/C) activation.

The online version of this article includes the following source data and figure supplement(s) for figure 4:

**Source data 1.** Uncropped blots for *Figure 4A, C–G*.

**Source data 2.** Raw data of Slp1 level measurement and time-course analyses of Cdc13-GFP at spindle pole body (SPB) for *Figure 4G, H*.

**Source data 3.** Raw unedited blots for *Figure 4*.

**Figure supplement 1.** Identification of in vivo phosphorylated or ubiquitylated residues in Slp1.

**Figure supplement 1—source data 1.** Uncropped gels for *Figure 4—figure supplement 1A*.

**Figure supplement 1—source data 2.** Full raw unedited Coomassie gel (*wild-type*) for *Figure 4—figure supplement 1A*.

**Figure supplement 1—source data 3.** Full raw unedited Coomassie gel (*pmk1Δ, pek1$^{DD}$*) for *Figure 4—figure supplement 1A*.

**Figure supplement 2.** MS spectra from mass spectrometric analyses of Slp1.

**Figure supplement 3.** Analysis of Slp1$^{Cdc20}$ phosphorylation at T480 in vivo upon forced tethering of Slp1$^{Cdc20}$ to Pmk1.

**Figure supplement 3—source data 1.** Uncropped blots for *Figure 4—figure supplement 3C*.

**Figure supplement 3—source data 2.** Raw data of quantitative analysis of Slp1 T480 phosphorylation levels for *Figure 4—figure supplement 3C*.

**Figure supplement 3—source data 3.** Full raw unedited blot (bead-bound, anti-pT480) for *Figure 4—figure supplement 3C*.

**Figure supplement 3—source data 4.** Full raw unedited blot (bead-bound, anti-GST) for *Figure 4—figure supplement 3C*.

**Figure supplement 3—source data 5.** Full raw unedited blot (input, anti-GST) for *Figure 4—figure supplement 3C*.

**Figure supplement 3—source data 6.** Full raw unedited blot (input, Cdc2) for *Figure 4—figure supplement 3C*.

**Figure supplement 4.** Sequence alignment performed with N- and C-terminal tails of *S. pombe* Slp1 and its homolog sequences from human (*H. sapiens*), frog (*X. laevis*), worm (*C. elegans*), and budding yeast (*S. cerevisiae*).

*Figure 4 continued on next page*

*Figure 4 continued*

**Figure supplement 5.** Mutant proteins of Slp1(T480A) and Slp1(S76A;T480A) are also stabilized in asynchronously growing cells.

**Figure supplement 5—source data 1.** Uncropped blots for *Figure 4—figure supplement 5*.

**Figure supplement 5—source data 2.** Raw data of Slp1 level measurement for *Figure 4—figure supplement 5*.

**Figure supplement 5—source data 3.** Full raw unedited blot (Slp1) for *Figure 4—figure supplement 5*.

**Figure supplement 5—source data 4.** Full raw unedited blot (Cdc2) for *Figure 4—figure supplement 5*.

**Figure supplement 6.** Time-course analyses and quantification of spindle checkpoint inactivation rate in phospho-deficient *slp1* mutants at 0 and 50 min after release at 30°C from *nda3*-mediated arrest.

**Figure supplement 6—source data 1.** Raw data of time-course analyses of Cdc13-GFP at spindle pole body (SPB) for *Figure 4—figure supplement 6A and B*.

Although Ser76 was identified as a Pmk1-dependent phosphorylation residue in our mass spectrometry analysis, it was considered as a Cdk1 targeting site in an earlier study (*Swaffer et al., 2018*). It is fairly possible that phosphorylation of Ser76 is promoted by both Cdk1 and Pmk1, which is conceivable as CDKs and MAPKs recognize the same core motif (SP/TP) (*Pinna and Ruzzene, 1996*), and crosstalk between CDK and MAPK has recently been reported for pheromone signaling in budding yeast (*Repetto et al., 2018*). We tested how the double phosphorylation sites (S76A;T480A) mutant affects Slp1$^{Cdc20}$ levels or APC/C activity, and observed that the protein levels of the Slp1$^{S76A;T480A}$ were further elevated compared to that of the Slp1$^{T480A}$ mutant while SAC activation efficiency was lowered (*Figure 4G, H*). Notably, Slp1$^{T480A}$ and Slp1$^{S76A;T480A}$ mutant proteins became easily detected in interphase cells (*Figure 4—figure supplement 5*), which is very different from wild-type Slp1 that normally accumulates exclusively in mitosis (*Yamada et al., 2000*).

We further tested whether replacement of other identified phospho-serine/threonine residues with alanines affects Slp1$^{Cdc20}$ protein levels and APC/C activity. We constructed strains carrying combinations of mutations of S28A, T31A, and S59A with S76A, and also with confirmed Pmk1 phosphorylation-deficient T480A, and found that mutations of S28A, T31A, and S59A did not affect Slp1$^{Cdc20}$ protein levels, though they could also profoundly compromise SAC activation efficiency similarly as the S76A;T480A mutant (*Figure 4G, H*, *Figure 4—figure supplement 5*, and *Figure 4—figure supplement 6*). We noticed that the mutations of S28A, T31A, and S59A also abolished the elevated Slp1$^{Cdc20}$ protein levels conferred by S76A;T480A mutations (*Figure 4G* and *Figure 4—figure supplement 5*). Thus, phosphorylation of S28, T31, and S59 targeted by Cdk1 most likely influences APC/C activation through a distinct mechanism from Pmk1-mediated S76/T480 phosphorylation.

Taken together, our data demonstrated that activated Pmk1 phosphorylates Slp1$^{Cdc20}$ in vivo, which leads to lowered Slp1$^{Cdc20}$ levels, delayed SAC inactivation and APC/C activation as we observed in *pek1$^{DD}$* cells. Also, Pmk1 can join Cdk1 to enhance the phosphorylation at multi-sites of their shared substrate Slp1$^{Cdc20}$ and collaboratively dampen APC/C activity.

## Pmk1- but not Cdk1-mediated Slp1$^{Cdc20}$ phosphorylation promotes its ubiquitylation

Our above data support the notion that the direct binding of Slp1$^{Cdc20}$ by Pmk1 mediates its phosphorylation, which results in its subsequent rapid turnover. It has been shown in both human and fission yeast cells that Apc15, one of the conserved components of APC/C, promotes Cdc20/Slp1$^{Cdc20}$ autoubiquitylation and its turnover by APC/C (*Mansfeld et al., 2011*; *May et al., 2017*; *Sewart and Hauf, 2017*; *Uzunova et al., 2012*). The increased Slp1$^{Cdc20}$ levels in *pmk1Δ* cells (*Figure 2C*) is reminiscent of the previous observations in *apc15Δ* mutant, which also shows elevated levels of Slp1$^{Cdc20}$ (*May et al., 2017*; *Sewart and Hauf, 2017*). We reasoned that the absence of Apc15 may reverse the negative effect of *pek1$^{DD}$* on Slp1$^{Cdc20}$ levels and APC/C activation. As expected, we indeed observed that deletion of *apc15* recovered Slp1$^{Cdc20}$ levels in *pek1$^{DD}$* cells (*Figure 5A*), and lowered the percentage of cells with Cdc13-GFP retained on SPBs in *nda3-KM311 pek1$^{DD}$* cells after cold treatment (*Figure 5B* and *Figure 5—figure supplement 1*). These observations are consistent with previous reports showing the relative abundance between checkpoint proteins and Slp1$^{Cdc20}$ is an important determinant of checkpoint robustness (*Heinrich et al., 2013*) and *apc15Δ* mutant is spindle checkpoint defective (*May et al., 2017*).

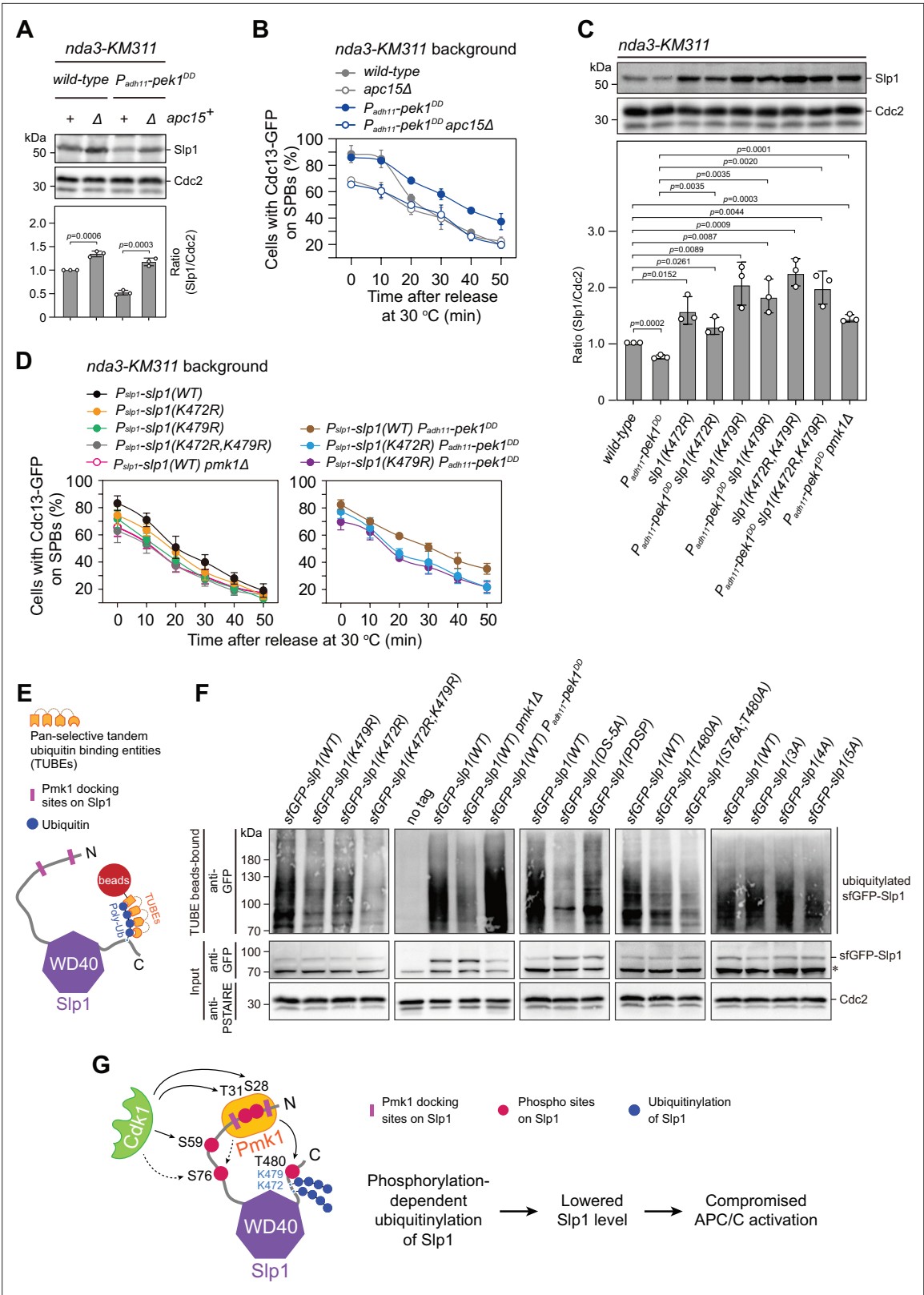

**Figure 5.** Pmk1- but not Cdk1-mediated Slp1[Cdc20] phosphorylation promotes its ubiquitylation. (**A**) Immunoblot analysis of Slp1[Cdc20] abundance in *apc15*[+]or *apc15Δ* background cells with indicated genotypes after being treated at 18°C for 6 hr. Slp1[Cdc20] levels were quantified as in *Figure 2B*. The experiment was repeated three times. The mean value for each sample was calculated, and p values were calculated against wild-type or *pek1[DD]* cells. (**B**) Time-course analyses of spindle assembly checkpoint (SAC) activation and inactivation in *nda3-KM311 cdc13-GFP* strains with indicated genotypes.

*Figure 5 continued on next page*

*Figure 5 continued*

The experiments were performed and analyzed as in *Figure 1D*. (C) Immunoblot analysis of Slp1$^{Cdc20}$ abundance in K472R, K479R, or K472R/K479R mutants. The experiment was repeated three times. The mean value for each sample was calculated, and p values were calculated against wild-type or *pek1$^{DD}$* cells. (D) Time-course analyses of SAC activation and inactivation in *nda3-KM311 cdc13-GFP* strains with K472R, K479R, or K472R/K479R mutations. The experiments were performed as in (B). (E) Schematic depiction of affinity pull-down assays using TUBE (tandem ubiquitin-binding entity) agarose beads to detect Slp1$^{Cdc20}$ ubiquitylation. (F) TUBE pull-down assays in *mts3-1* strains carrying sfGFP-tagged wild-type or mutants of Slp1$^{Cdc20}$ or *pmk1Δ* or *pek1$^{DD}$* mutations. The TUBE bead-bound samples were blotted with anti-GFP antibodies. Asterisk indicates unspecific bands recognized by anti-GFP antibodies. (G) Schematic summarizing the negative effect of the coupling of Pmk1 phosphorylation and K472/K479-mediated ubiquitylation on Slp1$^{Cdc20}$ abundance and anaphase-promoting complex/cyclosome (APC/C) activation.

The online version of this article includes the following source data and figure supplement(s) for figure 5:

**Source data 1.** Uncropped blots for *Figure 5A, C, F*.

**Source data 2.** Raw data of Slp1 level measurement and time-course analyses of Cdc13-GFP at spindle pole body (SPB) for *Figure 5A–D*.

**Source data 3.** Raw blots unedited for *Figure 5*.

**Figure supplement 1.** Quantification of spindle checkpoint inactivation rate in ubiquitylation-relevant *slp1* mutants at 0 and 50 min after release at 30°C from *nda3*-mediated arrest.

**Figure supplement 1—source data 1.** Raw data of time-course analyses of Cdc13-GFP at spindle pole body (SPB) for *Figure 5—figure supplement 1*.

**Figure supplement 2.** Slp1 proteins with mutations of K479R or K472R are also stabilized in asynchronously growing cells.

**Figure supplement 2—source data 1.** Uncropped blots for *Figure 5—figure supplement 2*.

**Figure supplement 2—source data 2.** Raw data of Slp1 level measurement for *Figure 5—figure supplement 2*.

**Figure supplement 2—source data 3.** Full raw unedited blot (Slp1) for *Figure 5—figure supplement 2*.

**Figure supplement 2—source data 4.** Full raw unedited blot (Cdc2) for *Figure 5—figure supplement 2*.

**Figure supplement 3.** Sequence alignment of C-terminal tails of Cdc20 homologs.

**Figure supplement 4.** Viability of *slp1(K472R;K479R)* cells is compromised.

**Figure supplement 5.** Enhanced Slp1$^{Cdc20}$ ubiquitylation in *pek1$^{DD}$* cells can be removed by Pmk1-docking-, Pmk1-phosphorylation- or ubiquitylation-deficient mutations in Slp1$^{Cdc20}$.

**Figure supplement 5—source data 1.** Uncropped blots for *Figure 5—figure supplement 5*.

**Figure supplement 5—source data 2.** Full raw unedited blot (bead-bound sfGFP-Slp1, blot 1) for *Figure 5—figure supplement 5*.

**Figure supplement 5—source data 3.** Raw unedited blots for *Figure 5—figure supplement 5*.

During the course of our mass spectrometry analysis on Slp1$^{Cdc20}$ phosphorylation, we also detected Lys 479 (K479) as an ubiquitin-modifying site (*Figure 4B*, *Figure 4—figure supplement 1*, and *Figure 4—figure supplement 2*). We tested whether K479 ubiquitylation is involved in regulating Slp1$^{Cdc20}$ protein stability. In agreement with the mass spectrometry analysis, mutation of K479 to arginine indeed increased the protein levels of Slp1$^{Cdc20}$ (*Figure 5C*), even in asynchronously growing cells (*Figure 5—figure supplement 2*), which is reminiscent of the stabilized Slp1$^{Cdc20}$ in *apc15Δ* mutant cells at interphase (*Sewart and Hauf, 2017*). These results indicated that K479 is likely Slp1$^{Cdc20}$ ubiquitylation relevant. Furthermore, *nda3-KM311 slp1$^{K479R}$* cells also demonstrated lowered metaphase arrest rate after cold treatment (*Figure 5D* and *Figure 5—figure supplement 1*), indicating defective SAC activation. Previous studies have identified two neighboring lysine residues close to the carboxy terminus of human Cdc20, Lys485 and Lys490, as the ubiquitylation sites in prometaphase cells, and mutating these two lysines to arginines completely prevented Cdc20 ubiquitylation (*Danielsen et al., 2011*; *Mansfeld et al., 2011*). Very interestingly, a second lysine residue, K472, is also present in fission yeast Slp1$^{Cdc20}$ adjacent to K479 (*Figure 4B* and *Figure 5—figure supplement 3*). Although K472 was not identified as an ubiquitin-modifying residue in our mass spectrometry analysis, replacement of K472 with arginine also efficiently stabilized Slp1$^{Cdc20}$, similar to K479R mutant (*Figure 5C* and *Figure 5—figure supplement 2*). Striking-ly, simultaneously mutating both K472 and K479 caused additive effects in elevating Slp1$^{Cdc20}$ protein levels and lowering SAC activation rate in wild-type or *pek1$^{DD}$* strain background (*Figure 5C, D*, *Figure 5—figure supplement 1*, and *Figure 5—figure supplement 2*), suggesting that likely both these two lysine residues at the C-terminal tail of Slp1$^{Cdc20}$ contribute to the protein turnover via ubiquitylation, which is very similar to the mechanism in human Cdc20. Consistently, using TUBEs (tandem ubiquitin-binding entities) (*Hjerpe et al., 2009*) to enrich for ubiquitinated proteins in metaphase-arrested *mts3-1* cells expressing sfGFP-Slp1, we were able to demonstrate that mutating both K472 and K479 largely removed the higher molecular weight

bands detected by immunoblotting, which corresponded to the ubiquitylated species of Slp1$^{Cdc20}$ (**Figure 5E, F**). We also noticed that Slp1$^{K472R; K479R}$ rendered more severely compromised viability than either of the single mutants (**Figure 5—figure supplement 4**), indicating blocking Slp1$^{Cdc20}$ ubiquitylation could bring about detrimental consequence to the cells.

Our observation that Slp1$^{Cdc20}$ levels were lowered in *pek1$^{DD}$* mutant (**Figure 2B**) suggested that attenuation of Slp1$^{Cdc20}$ levels upon constitutive activation of CIP signaling is very likely through facilitating its autoubiquitylation. To test this possibility, we compared the ubiquitylation levels of sfGFP-Slp1 in wild-type, *pmk1Δ* and *pek1$^{DD}$* cells by affinity pull-down assays using TUBEs (**Figure 5E**). Indeed, we observed a much weaker or stronger discrete laddering pattern of sfGFP-Slp1 in *pmk1Δ* or *pek1$^{DD}$* cells, respectively, than that of wild-type cells, indicating reduced or enhanced ubiquitylation levels (**Figure 5F**). Furthermore, mutations of T480A, S76A/T480A, and Pmk1-docking site, but not S28A/T31A- or S28A/T31A/S59A-containing mutants, also reduced polyubiquitylation of Slp1$^{Cdc20}$ in both *pek1$^+$* cells (**Figure 5F**) and *pek1$^{DD}$*-overexpressing (i.e. $P_{adh11}$-*pek1$^{DD}$*) cells (**Figure 5—figure supplement 5**). All these results demonstrated that Pmk1- but not Cdk1-mediated Slp1$^{Cdc20}$ phosphorylation promotes its ubiquitylation.

Together, these findings supported the idea that the CIP signaling pathway restrains APC/C activity through a cascade mechanism, in which Pmk1 directly binds to and phosphorylates Slp1$^{Cdc20}$ to facilitate its autoubiquitylation, and eventually lowers its levels (**Figure 5G**).

## Osmotic stress and cell wall damage trigger rapid Slp1$^{Cdc20}$ downregulation and mitotic exit delay

It has been well established that two MAPKs Sty1 and Pmk1 can be activated by environmental perturbations including osmotic stress, cell wall damage, thermal stress, and oxidative stress, among others (**Cansado et al., 2021**; **Madrid et al., 2006**; **Shiozaki and Russell, 1995b**). We wondered whether the phosphorylation of Slp1$^{Cdc20}$ by Pmk1 could tune the strength of SAC activation and APC/C activity in response to extracellular stress. To test this possibility, we performed the arrest-and-release experiments with *nda3-KM311* mutants and added 0.6 M KCl or 2 µg/ml of caspofungin 1 hr before release from metaphase arrest to elicit strong osmotic saline stress (KCl) or cell wall damage stress (caspofungin) response (**Madrid et al., 2006**; **Madrid et al., 2016**), respectively (**Figure 6A**). As previously reported, treatment with KCl or caspofungin resulted in full activation of both Pmk1 and Sty1 (for KCl treatment) or only Pmk1 (for caspofungin treatment) as determined using anti-phospho p42/44 or anti-phospho p38 antibodies, which serve as indicative of Pmk1 or Sty1 activation, respectively (**Figure 6B**). Intriguingly, treatment with KCl or caspofungin also correlated with a marked and rapid decrease in Slp1$^{Cdc20}$ levels (**Figure 6B**). It is noteworthy that SAC activation per se did not trigger activation of CIP pathway, as we did not observe any elevated Pmk1 phosphorylation in metaphase-arrested cells either by *nda3-KM311* mutation (see 'control' samples in **Figure 6B**) or by Mad2 over-expression (**Figure 6—figure supplement 1**).

Next, we investigated whether environmental perturbations affect APC/C–MCC interaction. Surprisingly, we did not see any alteration of APC/C–MCC interaction accompanying Pmk1 activation when cells were treated by caspofungin (**Figure 6C**), which was different from what we observed in $P_{adh11}$-*pek1$^{DD}$* cells (**Figure 2A**). Also, we failed to detect enhanced APC/C–MCC association as we observed in $P_{adh11}$-*wis1$^{DD}$* cells when treated by KCl to induce Sty1 activation (**Figure 6C**). The latter observation was most likely due to the disruptive effect of high concentration of KCl on protein–protein interactions, because we observed similar effect when we immunoprecipitated APC/C in the presence of 0.6 M KCl in the lysis buffer (**Figure 6C**, right panel). Currently, we could not exclude the possibility that activated Sty1 transiently enhanced APC/C–MCC association in yeast cells, which was then quickly counteracted by the disruptive effect of saline treatment. Nevertheless, the high SAC activation efficiency in mitotis-arrested cells exposed to osmotic stress (KCl) or cell wall damage stress (caspofungin) was greatly lowered by *pmk1* deletion (**Figure 6D** and **Figure 6—figure supplement 2**), indicating that Pmk1 could regulate the spindle checkpoint and APC/C activation under environmental stimuli. Furthermore, Pmk1-docking site mutant *slp1(DS-5A)* and Pmk1 phosphorylation-deficient mutants *slp1$^{T480A}$* and *slp1$^{S76A;T480A}$* could partially suppress the sustained SAC activation responding to osmotic and cell wall damage stresses (**Figure 6E** and **Figure 6—figure supplement 2**). Accordingly, protein levels of Pmk1 phosphorylation-deficient mutants Slp1$^{T480A}$ and Slp1$^{S76A;T480A}$ under above stresses were restored almost to the levels of wild-type Slp1 in untreated metaphase-arrested cells

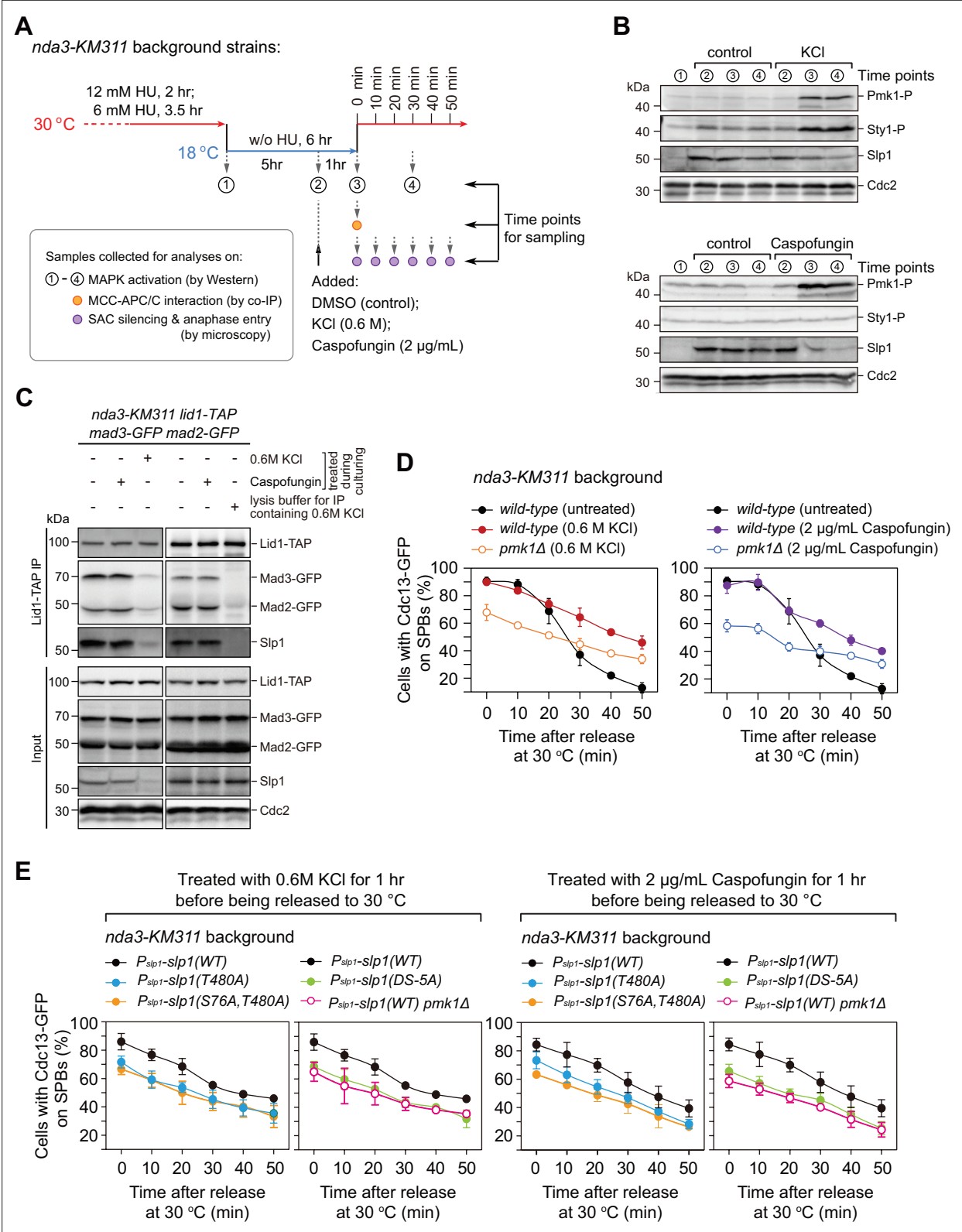

**Figure 6.** Osmotic stress and cell wall damage trigger rapid Pmk1 phosphorylation-dependent Slp1$^{Cdc20}$ downregulation and mitotic exit delay. (**A**) Schematic depiction of the experimental design for treatment with KCl or caspofungin during *nda3*-mediated spindle checkpoint activation to activate mitogen-activated protein kinases (MAPKs). Samples were collected at indicated time points for subsequent analyses including immunoblotting, co-immunoprecipitation (co-IP), and time-course analysis on spindle assembly checkpoint (SAC) or anaphase-promoting complex/

*Figure 6 continued on next page*

*Figure 6 continued*

cyclosome (APC/C) activation. (**B**) Immunoblot analysis of activation of MAPKs and Slp1[Cdc20] protein levels. Samples with or without indicated treatments were blotted with anti-phospho p42/44 and anti-phospho p38 antibodies as indicative of phosphorylated Pmk1 (Pmk1-P) or phosphorylated Sty1 (Sty1-P), respectively. Slp1[Cdc20] levels were detected with anti-Slp1 antibodies and anti-Cdc2 was used as loading control. (**C**) Co-immunoprecipitation analysis of APC/C–mitotic checkpoint complex (MCC) association upon environmental stress. Lid1-TAP was immunoprecipitated from *nda3-KM311*-arrested cells and associated Mad2-GFP, Mad3-GFP and Slp1[Cdc20] were detected as in *Figure 2A*. Note that APC/C–MCC association was disrupted when 0.6 M KCl was present during cell culturing or during immunoprecipitation procedures. (**D**) Time-course analyses of SAC activation and inactivation in *nda3-KM311 cdc13-GFP* strains with indicated genotypes after arrest at 18°C and KCl or caspofungin treatments. The experiment was repeated three times and the mean value and p value for each sample were calculated as in *Figure 1D*. (**E**) Time-course analyses of SAC activation and inactivation efficiency in Pmk1-docking- and phosphorylation-deficient *slp1* mutants under environmental stresses elicited by 0.6 M KCl or 2 µg/ml caspofungin.

The online version of this article includes the following source data and figure supplement(s) for figure 6:

**Source data 1.** Uncropped blots for *Figure 6B, C*.

**Source data 2.** Raw data of time-course analyses of Cdc13-GFP at spindle pole body (SPB) for *Figure 6D, E*.

**Source data 3.** Raw unedited blots for *Figure 6*.

**Figure supplement 1.** Activation of spindle assembly checkpoint (SAC) by Mad2 overexpression does not trigger Pmk1 activation.

**Figure supplement 1—source data 1.** Uncropped blots for *Figure 6—figure supplement 1*.

**Figure supplement 1—source data 2.** Full raw unedited blot (phosphorylated Pmk1) for *Figure 6—figure supplement 1*.

**Figure supplement 1—source data 3.** Full raw unedited blot (Slp1) for *Figure 6—figure supplement 1*.

**Figure supplement 1—source data 4.** Full raw unedited blot (Cdc2) for *Figure 6—figure supplement 1*.

**Figure supplement 2.** Quantification of spindle checkpoint inactivation rate in the presence of KCl or caspofungin treatment at 0 and 50 min after release at 30°C from *nda3*-mediated arrest.

**Figure supplement 2—source data 1.** Raw data of time-course analyses of Cdc13-GFP at spindle pole body (SPB) for *Figure 6—figure supplement 2*.

**Figure supplement 3.** Immunoblot analysis of activation of mitogen-activated protein kinases (MAPKs) and Slp1[Cdc20] protein levels in Pmk1 phosphorylation- and ubiquitylation-deficient *slp1* mutants under stress.

**Figure supplement 3—source data 1.** Uncropped blots for *Figure 6—figure supplement 3*.

**Figure supplement 3—source data 2.** Raw data of Slp1 level measurement for *Figure 6—figure supplement 3*.

**Figure supplement 3—source data 3.** Raw unedited blots for *Figure 6—figure supplement 3*.

---

(*Figure 6—figure supplement 3*). We also noticed that protein levels of ubiquitylation-deficient mutants, including Slp1[K472R], Slp1[K479R] and Slp1[K472R;K479R], were more efficiently maintained than Slp1[T480A] and therefore more resistant to osmotic and cell wall damage stresses (*Figure 6—figure supplement 3*).

## *pmk1Δ* cells are defective in faithful chromosome segregation upon environmental stress

Our above data demonstrated that, upon SAC activation, Pmk1 is involved in controlling the stability of Slp1[Cdc20] and thus the activation of APC/C under environmental stresses. We wondered whether the absence of *pmk1+* would affect the fidelity of chromosome segregation, particularly when cells are exposed to environmental stimuli. To answer this question, we treated $G_2$-arrested and -released *cdc25-22* cells with 0.6 M KCl or 2 µg/ml caspofungin to activate MAPKs (*Figure 7A*), and then examined possible chromosome segregation defects in *pmk1Δ* cells (*Figure 7B*). We found that stressed *pmk1Δ* cells displayed greatly increased frequency of lagging chromosomes and chromosome missegregation at mitotic anaphase compared to similarly treated wild-type cells or untreated *pmk1Δ* cells (*Figure 7C*). These data further validated the importance of Pmk1 in delaying anaphase onset and maintaining genome integrity under adverse stress.

## Discussion
## MAPKs are involved in regulation of mitotic anaphase onset in fission yeast

APC/C-mediated proteolysis of cyclin B and securin is key for anaphase entry by inactivating Cdk1 and permitting chromosome segregation, respectively. In response to unattached or tensionless

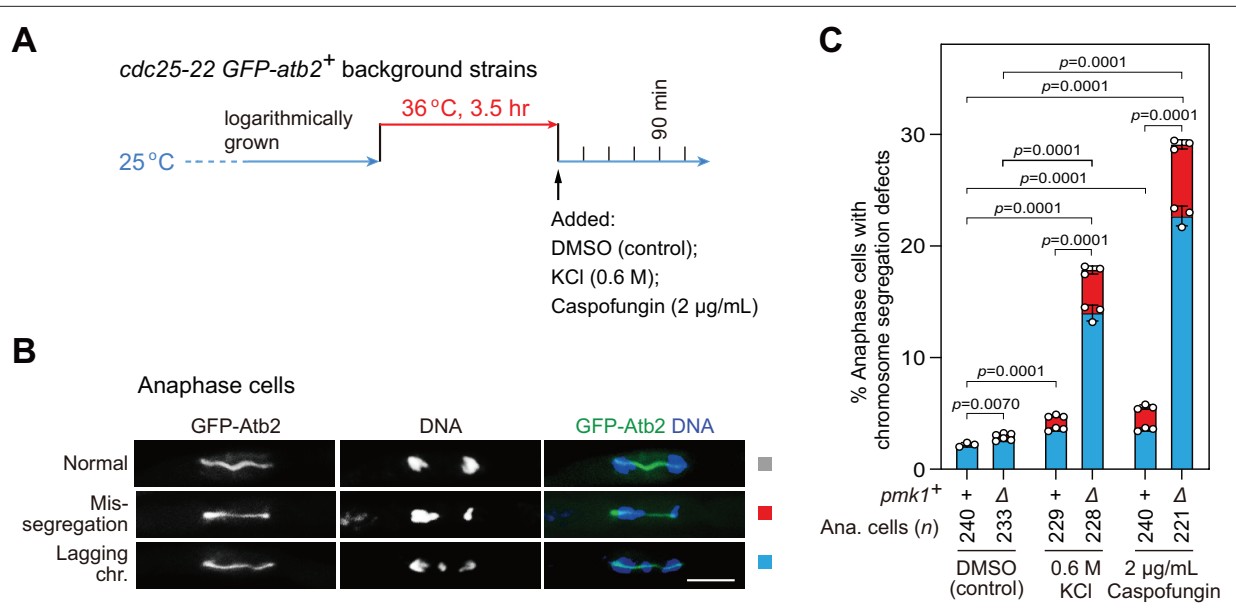

**Figure 7.** *pmk1Δ* cells are defective in faithful chromosome segregation upon environmental stress. (**A**) Schematic depiction of the experiment design for arrest-and-release of *cdc25-22 GFP-atb2*⁺ strains. Environmental stresses were imposed by 0.6 M KCl or 2 μg/ml caspofungin. Samples were taken at 90 min after being shifted back from 36 to 25°C to enrich anaphase cells. (**B, C**) Analyses of chromosome segregation in anaphase cells treated without or with KCl or caspofungin and after being fixed and stained with DAPI. Example pictures of anaphase cells with long spindles judged by GFP-Atb2 signals are shown (**B**). Anaphase cells with two categories of chromosome segregation defects (i.e. unequal chromosome segregation (mis-segregation) and lagging chromosomes) were quantified (**C**). >200 cells were counted for every sample. The mean values for each category were calculated, error bars indicate mean ± standard deviation of three independent experiments. p values were calculated with pooled data of two categories for each sample. *n*, numbers of anaphase cells analyzed. Scale bar, 5 μm.

The online version of this article includes the following source data for figure 7:

**Source data 1.** Raw data of defective chromosome segregation analyses for *Figure 7C*.

kinetochores, the SAC induces MCC generation. As a critical APC/C inhibitor, MCC fulfills its function through the physical binding of BubR1/Mad3 and Mad2 to two molecules of Cdc20/Slp1$^{Cdc20}$ (i.e. Cdc20$^{MCC}$ and Cdc20$^{APC/C}$), and this mechanism has been confirmed at least in both humans and fission yeast (*Alfieri et al., 2016*; *Chao et al., 2012*; *Izawa and Pines, 2015*; *Sewart and Hauf, 2017*; *Yamaguchi et al., 2016*). Thus, Cdc20 is unique amongst the MCC proteins in that it can function as either a co-activator or an inhibitor of the APC/C.

In the current study, we have uncovered a dual mechanism, in which two fission yeast MAPKs Pmk1 and Sty1 restrain APC/C activity through distinct manners, either by phosphorylating Slp1$^{Cdc20}$ to lower its levels or by phosphorylating an unknown substrate to promote MCC–APC/C association and thus SAC signaling strength, respectively (*Figure 8*). Therefore, our study has established another critical APC/C-inhibitory mechanism in the spindle checkpoint, though it will be important for future studies to identify the phosphorylation target(s) posed by SAP signaling pathway. We should mention that most of our experimental setups in this study were performed in cold-sensitive *nda3-KM311* mutants, which causes loss of both kinetochore–microtubule attachments and intra-kinetochore tension when being grown at 18°C. However, we were unable to distinguish whether MAPK involvement in SAC dynamics is relevant to only one or both perturbations. Nevertheless, our current study together with the earlier report *Edreira et al., 2020* have implicated Pmk1 in both the mitotic and cytokinesis surveillance in response to unfavorable environmental stimuli.

The regulation of mitotic entry by MAPK in fission yeast is triggered by many environmental insults, including high osmolarity, oxidative stress, heat shock, centrifugation, nutrient starvation, and rapamycin treatment (*Hartmuth and Petersen, 2009*; *Petersen and Hagan, 2005*; *Petersen and Nurse, 2007*; *Shiozaki and Russell, 1995a*; *Shiozaki et al., 1998*). We found that treatment of fission yeast cells with KCl or caspofungin, which provokes a strong osmotic stress or cell wall damage response and quick Pmk1 activation, correlated with attenuated Slp1$^{Cdc20}$ protein levels and delayed APC/C

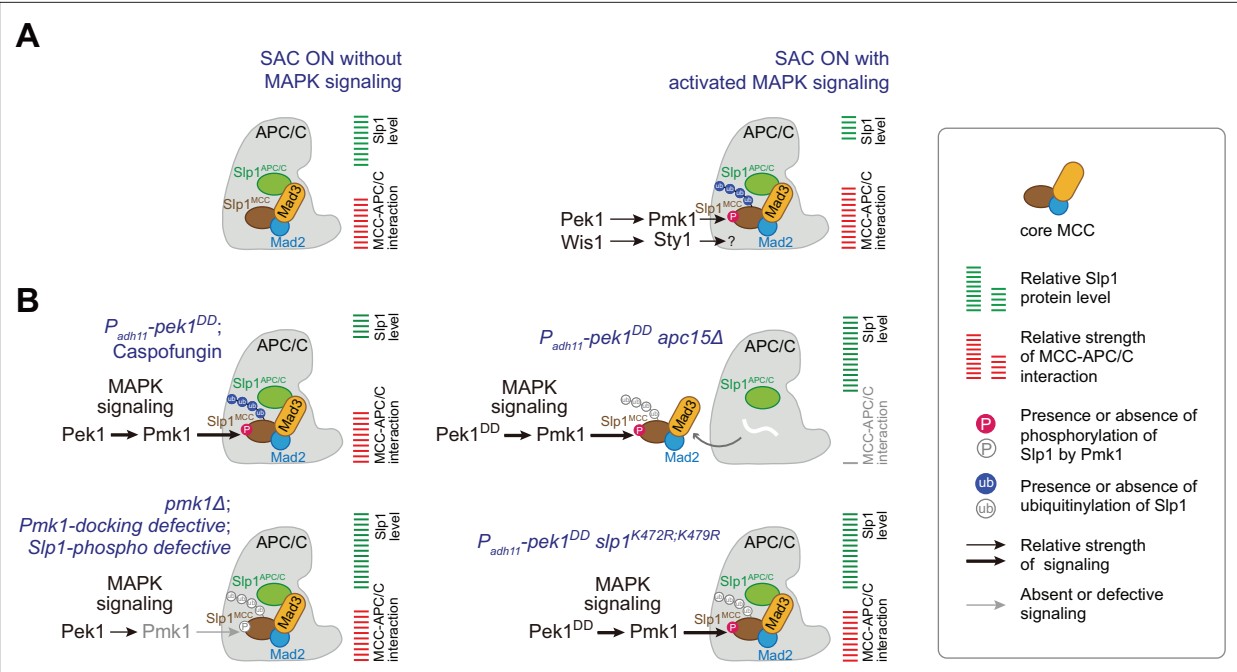

**Figure 8.** Summary of the mechanisms that how mitogen-activated protein kinases (MAPKs) negatively regulate APC/C activity in fission yeast. (**A**) Schematic depiction of a possible dual mechanism that how activated MAPK signaling pathways are involved in delaying APC/C activation and spindle assembly checkpoint (SAC) inactivation. Upon SAC and MAPK signaling activation, division of labor between the cell integrity pathway (CIP) and the stress-activated pathway (SAP) enables the phosphorylation of Slp1$^{Cdc20}$ and unidentified substrate(s) by Pmk1 and Sty1, respectively, which leads to lowered Slp1$^{Cdc20}$ level and enhanced mitotic checkpoint complex (MCC) affinity for APC/C. (**B**) Summary of the alteration of Slp1$^{Cdc20}$ protein levels and MCC–APC/C association strength under various conditions examined in this study. White wavy line indicates the absence of Apc15.

activation. Thus, our finding implicates MAPK Pmk1 as an important regulator for anaphase entry in response to adverse extracellular conditions. Intriguingly, one previous study showed that treatment of human cells with the carcinogen cadmium activates MAPK p38 (fission yeast Sty1 homolog) and triggers accelerated ubiquitination and proteolysis of Cdc20, which is essential for prometaphase arrest under SAC (*Yen and Yang, 2010*). Therefore, although different MAPKs might be employed to fulfil the task in yeast and higher eukaryotes, it is likely that the mechanisms regulating Cdc20 phosphorylation and protein levels by MAPKs are evolutionarily conserved in response to environmental stimuli.

## Phospho-regulation of Cdc20 by MAPK as a spindle assembly checkpoint-dependent APC/C-inhibitory mechanism

APC/C activity can be regulated at multiple levels. Cdk1-dependent APC/C phosphorylation is one of the major prerequisites for APC/C activation and anaphase onset, in which Apc3 and Apc1 are key targets for positive phospho-regulation (*Fujimitsu et al., 2016*; *Kraft et al., 2003*; *Qiao et al., 2016*; *Steen et al., 2008*; *Zhang et al., 2016*). However, it has also been shown that Cdk1-dependent phosphorylation of APC/C co-activator Cdc20 lessens its interaction with APC/C, thus serves as a negative phospho-regulation (*Labit et al., 2012*). In addition to its phosphorylation by Cdk1, Cdc20 can also be phosphorylated by multiple other kinases, such as Bub1, PKA, MAPK, and Plk1, which have been suggested in *Xenopus*, humans or budding yeast to be inhibitory to APC/C activation (*Jia et al., 2016* and reviewed in *Yu, 2007*). Therefore, Cdc20 could serve as a general integrator of multiple intracellular signaling cascades that regulate progression through mitosis.

Although multiple kinases have been proposed to be directly involved in phosphorylation of Cdc20 proteins (*Yu, 2007*), to identify precisely the in vivo phosphorylation sites in Cdc20 has turned out to be challenging, largely due to its protein scarcity and instability. Based on known Cdk1 core consensus (SP/TP), 9 residues in human Cdc20 were originally suggested as putative Cdk1 sites, but among them only Ser41 has been proven to be phosphorylated in vivo by a mass spectrometry analysis (*Kramer*

*et al., 2000*; *Yudkovsky et al., 2000*). However, subsequent mass spectrometry analyses identified Ser41 as one of the Bub1 phosphorylation sites both in vivo and in vitro and Thr106 as one of the Plk1 sites (*Jia et al., 2016*; *Tang et al., 2004*). It is very likely that there are not as many genuine Cdk1 sites in Cdc20 as originally assumed. One supporting evidence is that mutation of only one (Thr32) of the three putative Cdk sites to alanine in N terminus of *C. elegans* CDC-20 is sufficient to pheno-copy the 3A mutant for accelerated APC/C activation (*Kim et al., 2017*). Interestingly, Thr32 of *C. elegans* CDC-20 is conserved as Thr 59 and Thr68 in human and *Xenopus* Cdc20, respectively (see *Figure 4—figure supplement 4*). Based on two-dimensional tryptic phosphopeptide analysis, Thr68 in *Xenopus* Cdc20 has been regarded as an MAPK target site (*Chung and Chen, 2003*). These studies in diverse organisms raised an issue that whether multiple kinases share the same phosphorylation sites in Cdc20? It would not be surprising if it turns out to be the case, at least for Cdk1 and MAPKs, because both of them recognize the same core motif (SP/TP) (*Pinna and Ruzzene, 1996*).

By purifying Apc15-aasociated Slp1$^{Cdc20}$ proteins from checkpoint-arrested *pek1$^{DD}$* and *pmk1Δ* cells, we were able to identify Thr480 and Ser76 as the Pmk1 phosphorylation sites in vivo and three Pmk1-independent phosphorylation sites (S28, T31, and S59) based on mass spectrometry analyses. All these identified sites except Thr480 are located in the N terminal disordered portion of Slp1$^{Cdc20}$. This is similar to the phosphorylation residues identified by biochemical and functional analyses in other organisms, in which the major phosphorylation sites all fall within the N-terminus of Cdc20 homologs. In addition, we provided evidence that S28/T31 and T480 are phosphorylated by Cdk1 and Pmk1, respectively, both in vitro and in vivo. We noticed that there is a tendency for reported phosphorylation residues to be concentrated in the N terminus of Cdc20 proteins (*Jia et al., 2016*; *Tang et al., 2004*; *Yu, 2007*). However, due to the lack of clear sequence homology in their N- and C-terminal tails between Slp1$^{Cdc20}$ and its homologs in higher eukaryotes (see *Figure 4—figure supplement 4*), it is difficult to certainly assign the identified sites from different organisms as homologous sites. But it is possible that multiple kinases modify shared sites in Cdc20, this collaborative effect may be especially helpful under adverse environmental stress. Interestingly, currently all the reported Cdc20 phosphorylation events lead to inhibitory effects on APC/C no matter what kinases are involved (*Chung and Chen, 2003*; *D'Angiolella et al., 2003*; *Jia et al., 2016*; *Kim et al., 2017*; *Kramer et al., 2000*; *Labit et al., 2012*; *Tang et al., 2004*; *Yu, 2007*; *Yudkovsky et al., 2000*).

## Coupling of phosphorylation and autoubiquitylation of Slp1$^{Cdc20}$

The current model of inhibitory signaling from Cdc20 posits that its phosphorylation either promotes the formation of MCC or inhibits APC/C$^{Cdc20}$ catalytically, both of which are required for proper spindle checkpoint signaling (*Jia et al., 2016*; *Labit et al., 2012*; *Yamano, 2019*). Interestingly, our current study suggests Slp1$^{Cdc20}$ phosphorylation by fission yeast MAPK Pmk1 may represent another critical APC/C-inhibitory mechanism in the spindle checkpoint through reducing the protein levels of Slp1$^{Cdc20}$ via ubiquitylation-mediated degradation (*Figure 8*). This mechanism is distinct from that reported in *Xenopus* egg extracts and tadpole cells, in which phosphorylation of Cdc20 by MAPK is instead responsible for facilitated Cdc20 binding by MCC to prevent it from activating the APC/C (*Chung and Chen, 2003*). Actually, previous studies in fission and budding yeast have underlined the importance of relative abundance between checkpoint proteins and Slp1$^{Cdc20}$/Cdc20, which sets an important determinant of checkpoint robustness (*Heinrich et al., 2013*; *Pan and Chen, 2004*). A benefit of allowing the association of MCC-bound Cdc20 with APC/C is to provide an opportunity for Cdc20 autoubiquitylation by APC/C, a process aided by APC/C subunit Apc15 (*Mansfeld et al., 2011*; *May et al., 2017*; *Sewart and Hauf, 2017*; *Uzunova et al., 2012*). Notably, the depletion of *apc15$^+$* restores Slp1$^{Cdc20}$ levels in *pek1$^{DD}$* cells (*Figure 5A*), consistent with the idea that Pmk1-mediated Slp1$^{Cdc20}$ phosphorylation very likely facilitates its autoubiquitylation. Indeed, our biochemical and functional analyses further establish K479 and K472 in Slp1$^{Cdc20}$ as two key residues directly involved in its ubiquitylation, which likely follows its phosphorylation, particularly at adjacent Thr480. Despite K485 and K490 are similarly located at C terminus of human Cdc20 and can be ubiquitylated and responsible for Cdc20 degradation (*Danielsen et al., 2011*; *Mansfeld et al., 2011*), there have been no reports on phosphorylation-modified residues in this region. However, although the sequence homology within N- and C-terminal regions between Slp1$^{Cdc20}$ and Cdc20 homologs in higher eukaryotes is extremely low, we do notice that there are several lysine-flanking serine or threonine residues within Cdc20 C-terminal tails of higher eukaryotes (see *Figure 5—figure supplement 3*), this raises a

possibility that this region may be involved in phosphorylation-coupled ubiquitylation. Currently, we do not know how T480 phosphorylation accelerate ubiquitylation of K479 and K472 in Slp1$^{Cdc20}$. It is possible that phosphorylation of Slp1$^{Cdc20}$ might alter the mode of Cdc20 binding to APC/C which becomes conducive to catalyze Slp1$^{Cdc20}$ ubiquitylation.

In summary, our studies revealed a distinct and dual mechanism of MAPK-dependent anaphase onset delay imposed directly on APC/C co-activator Slp1$^{Cdc20}$ and an unidentified substrate, which may emerge as an important regulatory layer to fine tune APC/C and MCC activity upon SAC activation. Importantly, although this mechanism of APC/C regulation by two MAPKs has not been examined in other organisms, it is very likely widespread across eukaryotes including humans, given the involvement of MAPKs in general response to wide range of adverse environmental situations and similar regulations of APC/C and SAC in eukaryotes characterized so far. It is also plausible to assume that the MAPK-dependent APC/C inhibition pathway may contribute to the pathogenesis of multiple cancers. Indeed, Cdc20 upregulation has been demonstrated in several types of cancers, including breast and colon cancer, and it contributes to aggressive tumor progression and poor prognosis in gastric cancer and primary non-small cell lung cancer (*Chi et al., 2019*). One very recent study conducted in human pancreatic ductal adenocarcinoma cell lines revealed that inhibition of extracellular signal-regulated kinase (ERK) led to dephosphorylation of Cdc20 and concomitant loss of APC/C target proteins securin and cyclin B (CCNB1 and CCNB2) (*Klomp et al., 2024*). Because human ERKs are homologs of fission yeast Pmk1, the finding in this study supports a notion that very likely human MAPKs also negatively regulates APC/C through Cdc20 phosphorylation and the altered levels of Cdc20 in cancers are MAPK relevant. Nevertheless, inhibition of APC/C$^{Cdc20}$ activity or induction of Cdc20 degradation have already been listed as alternative therapeutic strategies to control cancer (*Greil et al., 2022*; *Jeong et al., 2022*). Thus, although the detailed molecular mechanisms may vary, further studies on MAPK-mediated APC/C$^{Cdc20}$ inhibition mechanism in human would lead to better understanding of its oncogenic role in tumor progression.

## Materials and methods
### Fission yeast strains, media, and genetic methods

Fission yeast cells were grown in either YE (yeast extract) rich medium or EMM (Edinburgh minimal medium) containing the necessary supplements. G418 disulfate (Sigma-Aldrich; A1720), hygromycin B (Sangon Biotech; A600230), or nourseothiricin (clonNAT; Werner BioAgents; CAS#96736-11-7) was used at a final concentration of 100 µg/ml and TBZ (Sigma-Aldrich; T8904) at 5–15 µg/ml in YE media where appropriate. For serial dilution spot assays, 10-fold dilutions of a mid-log-phase culture were plated on the indicated media and grown for 3–5 days at indicated temperatures.

To create strain with an ectopic copy of *slp1$^+$* at *lys1$^+$* locus on chromosome 1, the open reading frame of *slp1$^+$* (1467 bp) and its upstream 1504 bp sequence was first cloned into the vector pUC119-$P_{adh21}$-MCS-*hphMX6*-*lys1**  with *adh21* promoter ($P_{adh21}$) removed using the 'T-type' enzyme-free cloning method (*Chen et al., 2017*). The sequence of *hphMX6* in above plasmid was replaced by fragment corresponding to nourseothiricin resistance gene sequence by *Bgl*II–*Eco*RI cut and re-ligation, generating pUC119-$P_{slp1}$-*slp1$^+$*-$T_{adh1}$::*natMX6*-*lys1**. The resultant plasmids were linearized by *Apa*I and integrated into the *lys1$^+$* locus, generating the strains *lys1Δ::$P_{slp1}$-slp1$^+$-$T_{adh1}$::hphMX6* and *lys1Δ::$P_{slp1}$-slp1$^+$-$T_{adh1}$::nat$^R$*.

To generate the mutant strains harboring lysine/arginine (K/R) or serine/threonine to glutamic acid (E) or alanine (A) mutations, or lysine (K) to arginine (R) mutations within Slp1, the desired mutations were introduced into pUC119-$P_{slp1}$-*slp1$^+$*-$T_{adh1}$::*hphMX6* using standard methods of site-directed mutagenesis.

To generate the strains carrying superfolder GFP (sfGFP)-tagged wild-type or mutant Slp1, a fragment corresponding to the sequence of *sfGFP* was inserted in front of *slp1* in pUC119-$P_{slp1}$-*slp1*-$T_{adh1}$::*hphMX6*-*lys1** using the 'T-type' enzyme-free cloning method. The resulting plasmids were linearized and integrated into the *lys1$^+$* locus of chromosome 1 using the *hyg$^r$* marker. Then endogenous *slp1$^+$* was deleted and replaced with *ura4$^+$*.

To generate the strains expressing *ade6::$P_{adh1}$-GST-slp1$^{(456–488aa)}$::nat$^R$*, C-terminal sequence of *slp1$^+$* corresponding to 456–488aa was first cloned into the vector pUC119-$P_{adh1}$-MCS-*natMX6*-*lys1** and the sequence of *lys1** was replaced with *ade6$^+$* from pAde6-NotI-hphMX (a kind gift of Li-lin Du),

and then the sequence of GST from pGEX-4T-1 was inserted in front of *slp1*. The resulting plasmid pUC119-$P_{adh1}$-GST-slp1$^{(456-488aa)}$-natMX6-ade6 was linearized and integrated into the *ade6$^+$* locus of chromosome III using the *nat$^r$* marker.

To generate the strains expressing *ade6::P$_{adh1}$-slp1$^{(1-60aa)}$-mEGFP-2xNLS-GST-slp1$^{(456-488aa)}$::nat$^R$*, sequences corresponding to *slp1$^{(1-60aa)}$*, *mEGFP* and two tandem SV40 NLS (CCT AAG AAA AAA CGA AAA GTT GAG GAT CCT AAA AAG AAA CGA AAA GTT GAT) were cloned into the plasmid pUC119-$P_{adh1}$-GST-slp1$^{(456-488aa)}$-natMX6-ade6 using the 'T-type' enzyme-free cloning method. The resultant plasmid was linearized and integrated into the *ade6$^+$* locus as describes above.

To generate the strains expressing *lys1Δ::P$_{adh11}$-pmk1-GBP-mCherry::hyg$^R$*, the coding sequence of *pmk1$^+$* was first cloned into the vector pUC119-$P_{adh11}$-GBP-mCherry-hphMX6-lys1* (*Chen et al., 2017*), then the plasmid was linearized and integrated into the *lys1$^+$* locus in the genome.

To generate the constitutively active MAPKK strains *pek1$^{DD}$* or *wis1$^{DD}$*, the Ser234 and Thr238 of *pek1$^+$* in pUC119-$P_{adh21}$-6HA-pek1$^+$-$T_{adh1}$-hphMX6-lys1*, pUC119-$P_{adh11}$-6HA-pek1$^+$-$T_{adh1}$-hphMX6-lys1* and pUC119-$P_{adh11}$-6HA-pek1$^+$-$T_{adh1}$-kanMX6-ura4$^+$ and the Ser469 and Thr473 of *wis1$^+$* in pUC119-$P_{adh21}$-6HA-wis1$^+$-$T_{adh1}$-hphMX6-lys1*, pUC119-$P_{adh11}$-6HA-wis1$^+$-$T_{adh1}$-hphMX6-lys1* and pUC119-$P_{adh11}$-6HA-wis1$^+$-$T_{adh1}$-kanMX6-ura4$^+$ were changed to aspartic acids (D) by standard methods of site-directed mutagenesis. The resulting plasmids were linearized and integrated into the *lys1$^+$* locus of chromosome I or the *ura4$^+$* locus of chromosome III using the *kan$^r$* marker. A list of the yeast strains used is in *Supplementary file 1*.

## Fission yeast cell synchronization methods

For *nda3-KM311* strains, cells were grown at the permissive temperature for *nda3-KM311* (30°C) to mid-log phase, synchronized at S phase by adding HU (Sangon Biotech; A600528) to a final concentration of 12 mM for 2 hr followed by a second dose of HU (6 mM final concentration) for 3.5 hr. HU was then washed out and cells were released at specific temperatures as required by subsequent experiments.

For *cdc25-22* background strains, cells were first grown at 25°C and then arrested at the $G_2$/M transition by shifting to 36°C for 3.5 hr. To allow cells to progress into mitosis, the cultures were released by shifting back to 25°C. Progression through cell cycle was monitored by DAPI staining after being fixed and counting binucleate cells.

## Spindle checkpoint activation and silencing assays in fission yeast

For checkpoint silencing assay in the absence of microtubules, mid-log *cdc13-GFP nda3-KM311* cells were first synchronized with HU at 30°C and then arrested in early mitosis by shifting to 18°C for 6 hr, followed by incubation at 30°C to allow spindle reformation and therefore spindle checkpoint inactivation. For osmotic stress or cell wall damage stress experiments, 0.6 M KCl or 2 μg/ml caspofungin was added into cultures during 18°C incubation 1 hr before being shifted to 30°C. Cells were withdrawn at certain time intervals and fixed with cold methanol and stained with DAPI. 200–300 cells were analyzed for each time point. Each experiment was repeated at least three times.

To achieve spindle checkpoint activation and metaphase arrest through a different mechanism, Mad2 was overexpressed from the *nmt1* promotor (i.e. $P_{nmt1}$). To induce Mad2 overexpression, strains carrying *P$_{nmt1}$-mad2::leu1$^+$* were pre-cultured in minimal medium with supplements (EMM5S) and 15 μM thiamine, cells were washed five times in EMM5S to remove thiamine before being grown in EMM5S at 30°C for 18 hr. Mitotic arrest was determined by DAPI staining and GFP-Atb2-labeled short spindles.

## Immunoblotting and immunoprecipitation

For routine western blot and immunoprecipitation experiments, yeast cells were collected from *nda3-KM311* arrested and released cultures, followed by lysing with glass bead disruption using Bioprep-24 homogenizer (ALLSHENG Instruments, Hangzhou, China) in NP-40 lysis buffer (6 mM $Na_2HPO_4$, 4 mM $NaH_2PO_4$, 1% NP-40, 150 mM NaCl, 2 mM ethylene diamine tetraacetic acid (EDTA), 50 mM NaF, 0.1 mM $Na_3VO_4$) plus protease inhibitors as previously described (*Wang et al., 2012*). Proteins were immunoprecipitated by IgG Sepharose beads (GE Healthcare; 17-0969-01) (for Lid1-TAP) or anti-GFP nanobody agarose beads (AlpalifeBio, Shenzhen, China; KTSM1301) (for Mad3-GFP). Immunoblot

analysis of cell lysates and immunoprecipitates was performed using appropriate antibodies at 1:500 to 1:5000 dilutions and was read out using chemiluminescence.

## Detection of in vivo phosphorylation of Slp1[Cdc20] at Thr480 and Ser28/Thr31

For detection of phosphorylation of Slp1[Cdc20] at Thr480 in vivo, *mts3-1* mutant strains expressing *ade6::P*$_{adh1}$*-GST-slp1(456–488aa)::natR* in wild-type, *pmk1Δ* and *lys1Δ::P*$_{adh11}$*-6xHA-pek1*$^{DD}$ background were arrested at 36°C for 3.5 hr, followed by pull-down of GST-Slp1(456–488aa) by glutathione Sepharose 4B (GE Healthcare). Purified GST-Slp1(456–488aa) was detected with either goat anti-GST horseradish peroxidase (HRP)-conjugated antibodies (GE Healthcare) or rabbit polyclonal anti-pThr480 antibodies.

For detection of phosphorylation of Slp1[Cdc20] at Ser28/Thr31 in vivo, *nda3-KM311 apc15-13myc::kanR* strains expressing *slp1*$^+$ or *slp1(S28A;T31A)* in wild-type or *cdc2-asM17* background were first synchronized with HU at 30°C and then arrested in early mitosis by shifting to 18°C for 6 hr. For *cdc2-asM17* strains, 1 μM 1-NM-PP1 (Toronto Research Chemicals; A603003) was added into one of the cultures to inactivate Cdc2 (Cdk1) 20 min before being collected. Apc15-13myc was immunoprecipitated by anti-c-Myc magnetic beads (MedChemExpress (MCE); HY-K0206). To remove Slp1[Cdc20] phosphorylation, one of the immunoprecipitated samples was treated with 400 units of Lambda protein phosphatase (New England Biolabs, P0753S) at 30°C for 30 min. Co-immunoprecipitated Slp1[Cdc20] was detected with rabbit polyclonal anti-Slp1 antibodies, and the phosphorylated form of Slp1[Cdc20] was detected with rabbit polyclonal anti-pSer28/pThr31 antibodies, respectively.

## Purification of APC/C and analyses by mass spectrometry

To prepare APC/C for mass spectrometry analyses, Apc15-GFP was purified from 6 l cultures of *nda3-KM311* arrested cells with desired genetic background. Cells were disrupted and cell lysates were prepared as described above for routine immunoprecipitation experiments.

For mass spectrometry analyses, purified samples were first run on PAGE gels, after staining of gels with Coomassie blue, excised gel segments were subjected to in-gel trypsin (Promega, V5111) digestion and dried. Samples were then analyzed on a nanoElute (plug-in V1.1.0.27; Bruker, Bremen, Germany) coupled to a timsTOF Pro (Bruker, Bremen, Germany) equipped with a CaptiveSpray source. Peptides were separated on a 15 cm × 75 μm analytical column, 1.6 μm C18 beads with a packed emitter tip (IonOpticks, Australia). The column temperature was maintained at 55°C using an integrated column oven (Sonation GmbH, Germany). The column was equilibrated using 4 column volumes before loading sample in 100% buffer A (99.9% MilliQ water, 0.1% formic acid (FA)) (both steps performed at 980 bar). Samples were separated at 400 nl/min using a linear gradient from 2% to 25% buffer B (99.9% acetonitrile (ACN), 0.1% FA) over 90 min before ramping to 37% buffer B (10 min), ramp to 80% buffer B (10 min) and sustained for 10 min (total separation method time 120 min). The timsTOF Pro (Bruker, Bremen, Germany) was operated in parallel accumulation-serial fragmentation (PASEF) mode using Compass Hystar 6.0. Mass range 100–1700 *m/z*, 1/K0 Start 0.6 V·s/cm² End 1.6 V·s/cm², Ramp time 110.1 ms, Lock Duty Cycle to 100%, Capillary Voltage 1600 V, Dry Gas 3 l/min, Dry Temp 180°C, PASEF settings: 10 MS/MS scans (total cycle time 1.27 s), charge range 0–5, active exclusion for 0.4 min, Scheduling Target intensity 10,000, Intensity threshold 2500, CID collision energy 42 eV. All raw files were analyzed by PEAKS Studio Xpro software (Bioinformatics Solutions Inc, Waterloo, ON, Canada). Data were searched against the *S. pombe* proteome sequence database (Uniprot database with 5117 entries of protein sequences at https://www.uniprot.org/proteomes/UP000002485). De novo sequencing of peptides, database search and characterizing specific PTMs were used to analyze the raw data; false discovery rate was set to ≤1%, and [−10*log(p)] was calculated accordingly where p is the probability that an observed match is a random event. The PEAKS used the following parameters: (1) precursor ion mass tolerance, 20 ppm; (2) fragment ion mass tolerance, 0.05 Da (the error tolerance); (3) tryptic enzyme specificity with two missed cleavages allowed; (4) monoisotopic precursor mass and fragment ion mass; (5) a fixed modification of cysteine carbamidomethylation; and (6) variable modifications including *N*-acetylation of proteins and oxidation of Met.

## Ubiquitin affinity pull-down assays

For detection of in vivo ubiquitylation of Slp1$^{Cdc20}$, strains all carried *mts3-1* mutation were used. sfGFP-tagged wild-type Slp1 or Slp1 mutants were expressed in the genomically integrated form of $P_{slp1}$-sfGFP-slp1(WT or mutants)::hyg$^R$ in wild-type, *pmk1Δ* or $P_{adh11}$-6xHA-pek1$^{DD}$ background. Cultures were first grown at 25°C to mid-log phase and then shifted to 36°C for 3.5 hr to block cells in mitosis prior to harvesting and cell lysis. Ubiquitinated proteins were pulled down from yeast lysates using TUBEs (LifeSensors, UM402) according to the manufacturer's instructions. The bound proteins were eluted from washed Agarose-TUBE2 beads and analyzed by western blotting using a mouse monoclonal anti-GFP antibody (Beijing Ray Antibody Biotech; RM1008). Cell extracts were also immunoblotted for GFP and Cdc2 as loading controls (inputs).

## Expression and purification of recombinant proteins

GST or MBP fusion constructs of fission yeast full-length or truncated Slp1$^{Cdc20}$, Pmk1, Pek1, and Atf1 were generated by polymerase chain reaction (PCR) of the corresponding gene fragments from yeast genomic DNA or cDNA and cloned in frame into expression vectors pGEX-4T-1 (GE Healthcare) or pMAL-c2x (New England Biolabs), respectively. Site-directed mutagenesis was done by standard methods to generate vectors carrying Pek1-DD(S234D, T238D) or Slp1$^{Cdc20}$ harboring K(Lys)/R(Arg) to E(Glu) or A(Ala) mutations at individual or combined sites including K19, K20, R21, K47, R48, K140, K141, K161, R162, R163, K472, and R473. Integrity of cloned DNA was verified by sequencing analysis. All recombinant GST- or MBP-fusion proteins were expressed in *Escherichia coli* BL21 (DE3) cells. The recombinant proteins were purified by glutathione Sepharose 4B (GE Healthcare) or Amylose Resin High Flow (New England Biolabs), respectively, according to the manufacturers' instructions.

## In vitro pulldown assay

The recombinant MBP-Slp1$^{Cdc20}$ and MBP, or GST-Pmk1 and GST proteins were produced in bacteria and purified as described above. To detect the interaction between bacterially expressed Slp1$^{Cdc20}$ and Pmk1-HA expressed in yeast, about 1 μg of MBP-Slp1$^{Cdc20}$ or MBP immobilized on amylose resin was incubated with cleared cell lysates prepared from yeast strains *nda3-KM311 pmk1-6His-HA*, which were HU synchronized and arrested at prometaphase by being grown at 18°C for 6 hr. To detect the direct interaction between bacterially expressed Slp1 and Pmk1, about 1 μg of GST-Pmk1 or GST immobilized on glutathione Sepharose 4B resin was incubated with cleared bacterial lysates expressing MBP-Slp1$^{Cdc20}$. Samples were incubated for 1–3 hr at 4°C. Resins were washed three times with lysis buffer (6 mM Na$_2$HPO$_4$, 4 mM NaH$_2$PO$_4$, 1% NP-40, 150 mM NaCl, 2 mM EDTA, 50 mM NaF, 0.1 mM Na$_3$VO$_4$, plus protease inhibitors), suspended in SDS sample buffer, and then subject to SDS–PAGE electrophoresis and Coomassie brilliant blue staining.

## Non-radioactive in vitro kinase assay

GST-Slp1$^{Cdc20}$, GST-Atf1, GST-Pmk1, and GST-Pek1$^{DD}$ fusions were expressed and purified from *E. coli* with glutathione Sepharose 4B as described above. Pmk1-HA-His and Cdc13-GFP were purified from yeast lysates by immuprecipitation with anti-HA or anti-GFP antibodies, respectively.

For detection of phosphorylation of Slp1$^{Cdc20}$ and Atf1 by yeast-derived Pmk1, glutathione Sepharose 4B beads bound with GST-Slp1$^{Cdc20}$ or GST-Atf1 and agarose beads bound with Pmk1-HA-His were mixed and washed three times with kinase buffer (50 mM Tris–HCl, pH 7.5, 10 mM MgCl$_2$, 1 mM ethylene glycol tetraacetic acid (EGTA)), then the reactions were incubated in kinase buffer with 20 μM ATPγS (Sigma-Aldrich; A1388) at 30°C for 45 min. The kinase reactions were stopped by adding 20 mM EDTA, and the reaction mixture was alkylated after incubation at room temperature with 2.5 mM *p*-nitrobenzyl mesylate (Abcam; ab138910) for 1 hr. The alkylation reaction was stopped by boiling in SDS–PAGE loading buffer. Phosphorylated proteins were detected with rabbit monoclonal anti-Thiophosphate ester antibody (Abcam, ab239919).

For detection of phosphorylation of Slp1$^{Cdc20}$ at Thr480 by bacterially expressed Pmk1, beads bound with GST-Pmk1, GST-Pek1(DD) and fused substrates GST-Slp1$^{(456–488aa)}$ or GST-Slp1$^{(456–488aa, T480A)}$ were washed and incubated in kinase buffer (50 mM Tris–HCl, pH 7.5, 10 mM MgCl$_2$, 1 mM EGTA) with 10 mM ATP (Sangon Biotech; A600020-0005) at 30°C for 45 min. The reaction was stopped by boiling in SDS–PAGE loading buffer. Phosphorylated proteins were detected with rabbit polyclonal anti-pThr480 antibodies.

For detection of phosphorylation of Slp1[Cdc20] at Ser28/Thr31 by Cdk1, beads bound with bacterially expressed MBP-Slp1[(1–190aa)] or MBP-Slp1[(1–190aa,S28A;T31A)] and agarose beads bound with Cdc13-GFP/Cdc2 from yeast lysate were mixed and washed three times with kinase buffer (10 mM Tris–HCl, pH 7.4, 10 mM MgCl$_2$, 1 mM dithiothreitol (DTT)), then the reactions were incubated in kinase buffer with 10 mM ATP (Sangon Biotech; A600020-0005) at 30°C for 45 min. To inactivate Cdc2-as1, 5 µM 1-NM-PP1 (Toronto Research Chemicals; A603003) was added into the reaction. The reaction was stopped by boiling in SDS–PAGE loading buffer. Phosphorylated proteins were detected with rabbit polyclonal anti-pSer28/pThr31 antibodies.

## Antibodies for immunoblotting

The phospho-site-specific antibodies were made in an in-house facility (for anti-Slp1-pS28/pT31) or by ABclonal Technology Co, Ltd (Wuhan, China) (for anti-Slp1-pT480) by immunizing rabbits with pS28/pT31- or pT480-containing peptides (LVFPN(S-p)PI(T-p)PLHQQ or PITK(T-p)PSSS, respectively) coupled to haemocyanin (Sigma). The purified antibodies were concentrated to 0.2 mg/ml and used at 1:100 dilution. The following antibodies used for immunoblot analyses were purchased from the indicated commercial sources and were used at the indicated dilution: peroxidase–anti-peroxidase soluble complex (Sigma-Aldrich; P1291; 1:10,000); rabbit polyclonal anti-Myc (GeneScript; A00172-40; 1:2000); mouse monoclonal anti-GFP (Beijing Ray Antibody Biotech; RM1008; 1:2000); rat monoclonal anti-HA (Roche, Cat. No. 11 867 423 001; 1:2000); rabbit polyclonal anti-Slp1 (*Kim et al., 1998*) (1:500); rabbit polyclonal anti-phospho-p44/42 (detecting activated Pmk1) (Cell Signaling Technology; #9101; 1:1000); rabbit monoclonal anti-phospho-p38 (Cell Signaling Technology; #4511; 1:1000); rabbit polyclonal anti-PSTAIRE (detecting Cdc2) (Santa Cruz Biotechnology; sc-53; 1:1000); rabbit monoclonal anti-Thiophosphate ester antibody (Abcam; ab239919; 1:5000); goat anti-GST HRP-conjugated antibody (RRID:AB_771429; GE Healthcare; 1:10,000). Secondary antibodies used were goat anti-mouse or goat anti-rabbit polyclonal IgG (H+L) HRP conjugates (Thermo Fisher Scientific; #31430 or #32460; 1:10,000).

## RT-qPCR

For RT-qPCR of *slp1*[+] mRNA, *nda3-KM311* strains were grown, synchronized by HU and arrested at metaphase with SAC being activated as described above. Cells were collected and total cellular RNA was isolated by using TRIZOL method. cDNA was prepared using oligo(dT) and PrimeScript reverse transcriptase (Takara, D2680A). Real-time PCR was performed in the presence of SYBR Green using an Applied Biosystems 7900HT light cycler. Relative RNA levels were calculated using the *ΔCT* method and normalized to *act1*[+] levels.

## Fluorescence microscopy

Cdc13-GFP proteins were observed in cells after fixation with cold methanol. For DAPI staining of nuclei, cells were fixed with cold methanol, washed in PBS and resuspended in PBS plus 1 µg/ml DAPI. Photomicrographs of cells were obtained using a Nikon 80i fluorescence microscope coupled to a cooled CCD camera (Hamamatsu, ORCA-ER) or a Perkin Elmer spinning-disk confocal microscope (UltraVIEW VoX) with a 100× NA 1.49 TIRF oil immersion objective (Nikon) coupled to a cooled CCD camera (9100-50 EMCCD; Hamamatsu Photonics) and spinning disk head (CSU-X1, Yokogawa). Image processing and analysis were carried out using Element software (Nikon), ImageJ software (National Institutes of Health) and Adobe Photoshop.

## Statistical analyses and reproducibility

No statistical methods were used to predetermine sample size. Sample size was as large as practicable for all experiments. The experiments were randomized and the investigators were blinded to group allocation during experiments but not blinded to outcome assessment and data analysis.

All experiments were independently repeated two to more than three times with similar results obtained. For quantitative analyses of each time-course experiment, ≥300 cells were counted for each time point or sample. The same sample was not measured or counted repeatedly. No data were excluded from our studies. Data collection and statistical analyses were performed using Microsoft Office Excel or GraphPad Prism 6 softwares. Data are expressed as mean values with error bars

(± standard deviation/s.d.) and were compared using two-tailed Student's *t*-tests unless indicated otherwise.

## Acknowledgements

We thank Kathy Gould, Li-lin Du, Silke Hauf, Jonathan Millar, Takashi Toda, Yoshinori Watanabe, Elena Hidalgo and National BioResource Project (NBRP), Japan (http://yeast.nig.ac.jp/yeast/) for fission yeast strains or plasmids; Rafael Daga for communication on unpublished results; Jose Cansado for advice on active MAPK detection. We also thank Yaying Wu, Zheni Xu, and Chang-chuan Xie for mass spectrometry experiments and data analyses; Yu-ting He for contributing a few time-course experimental data; and Shiyu Huang and Jun-ming Zhuang for help with construction of some plasmids or yeast strains. This work was supported by grants from the National Natural Science Foundation of China (No. 32170731, No. 31671411) to QW Jin.

## Additional information

### Funding

| Funder | Grant reference number | Author |
| --- | --- | --- |
| National Natural Science Foundation of China | 32170731 | Quan-wen Jin |
| National Natural Science Foundation of China | 31671411 | Quan-wen Jin |

The funders had no role in study design, data collection, and interpretation, or the decision to submit the work for publication.

### Author contributions

Li Sun, Xuejin Chen, Shuang Bai, Data curation, Formal analysis, Investigation, Visualization, Methodology; Chunlin Song, Libo Liu, Data curation, Formal analysis, Validation, Investigation, Visualization, Methodology; Wenjing Shi, Data curation, Formal analysis, Validation, Investigation, Methodology; Xi Wang, Shuang-min Wang, Formal analysis, Investigation; Jiali Chen, Formal analysis, Validation, Investigation; Chengyu Jiang, Formal analysis, Investigation, Visualization; Zhou-qing Luo, Formal analysis, Writing – original draft; Ruiwen Wang, Formal analysis, Supervision, Investigation, Writing – review and editing; Yamei Wang, Conceptualization, Formal analysis, Supervision, Investigation, Writing – original draft, Project administration, Writing – review and editing; Quan-wen Jin, Conceptualization, Resources, Data curation, Formal analysis, Supervision, Funding acquisition, Visualization, Methodology, Writing – original draft, Project administration, Writing – review and editing

### Author ORCIDs

Quan-wen Jin ⓘ https://orcid.org/0000-0001-6146-6910

Reviewer #1 (Public review): https://doi.org/10.7554/eLife.97896.3.sa1
Reviewer #2 (Public review): https://doi.org/10.7554/eLife.97896.3.sa2
Author response https://doi.org/10.7554/eLife.97896.3.sa3

## Additional files

### Supplementary files
• MDAR checklist
• Supplementary file 1. Yeast strains used in this study.

### Data availability

The mass spectrometry proteomics data have been deposited to the ProteomeXchange Consortium via the PRIDE partner repository with the dataset identifier PXD048800. All data generated or

analyzed during this study are included in the manuscript and supporting files; source data files have been provided for Figures 1–7, Figure 1—figure supplements 1–3, Figure 2—figure supplement 1, Figure 3—figure supplements 2 and 3, Figure 4—figure supplements 1; 3; 5 and 6, Figure 5—figure supplements 1; 2 and 5, Figure 6—figure supplements 1–3.

The following dataset was generated:

| Author(s) | Year | Dataset title | Dataset URL | Database and Identifier |
|---|---|---|---|---|
| Wang X, Wang S, Wang Y, Jin Q | 2024 | Pmk1 kinase-dependent phosphosite mapping on Slp1 | https://www.ebi.ac.uk/pride/archive/projects/PXD048800 | PRIDE, PXD048800 |

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
