## [Editor Report · eLife assessment]

The regulation of mitosis and the dynamics of the mitotic spindle in it are central to cell division with high fidelity and crucial for normal division and development and defects therein can lead to disease. A key component of ensuring the fidelity is the "spindle assembly checkpoint". This **valuable** study using **convincing** experimental approaches in fission yeast has revealed novel links between the MAP-kinase signalling pathway modulating the spindle assembly checkpoint.

---

## [Referee Report · Reviewer #1 (Public review)]

Summary:

This manuscript addresses two main issues: (i) do MAPKs play an important role in SAC regulation in single cell organism such as *S pombe*? (ii) what is the nature of their involvement and what are their molecular targets?

The authors have extensively used the cold-sensitive β-tubulin mutant to activate or inactivate SAC employing an arrest-release protocol. Localization of Cdc13 (cyclin B) to the SPBs is used as a readout for the SAC activation or inactivation. The roles of two major MAPK pathways i.e. stress activated pathway (SAP) and cell integrity pathway (CIP), have been explored in this context (with CIP more extensively than SAP). Sty1Δ or pmk1Δ mutants were used to inactivate the SAP or CIP pathways and wis1DD or pek1DD expression was utilized to constitutively activate these pathways, respectively. Lowering of Slp1Cdc20 abundance (by phosphorylation of Slp1-Thr 480) is revealed as the main function of MAPK to augment the robustness of spindle assembly checkpoint.

Strengths:

The experiments are generally well-conducted, and the results support the interpretations in various sections. The experimental data clearly support some of the key conclusions:

(i) while inactivation of SAP and CIP compromises SAC-imposed arrest, their constitutive activation delays the release from the SAC-imposed arrest (ii) CIP signaling, but not SAP signaling, attenuates Slp1Cdc20 levels (iii) Pmk1 and Cdc20 physical interact and Pmk1-docking sequences in Slp1 (PDSS) is identifies and confirmed by mutational/substitution experiments (iv) Thr480 (and also S76) is identified as the residue phosphorylated by Pmk1. S28 and T31 are identified as Cdk1 phosphorylation sites. These are confirmed by mutational and other related analyses (v) Functional aspects of the phosphorylation sites have been elucidated to some extent: (a) Phosphorylation of Slp1-T480 by Pmk1 reduces its abundance thereby augmenting the SAC-induced arrest (b) S28, T31 (also S59) are phosphorylated by Cdk1 (v) K472 and K479 residues are involved in ubiquitylation of Slp1

Weaknesses:

(i) Cdc13 localization to SPBs has been used as a readout for SAC activation/inactivation throughout the manuscript. However, the only image showing such localization (Figure 1C) is of poor quality where the Cdc13 localization to SPBs barely visible. This should be replaced by a better image.

(ii) The overlapping error-bars in Cdc13-localization data in some figures (for instance Figure 3E and 4H) makes the effect of various mutations on SAC activation/inactivation rather marginal. In some of these cases, Western-blotting data support the author's conclusions better.

(iii) This specific point is not really a weakness but rather a loose end:

One of the conclusions of this study is that MAPK (PMK1) contributes to the robustness of SAC-induced arrest by lowering the abundance of Slp1Cdc20. The authors have used pmk1Δ or constitutively activating the MAPK pathways (Pek1DD) and documenting their effect on SAC activation/inactivation dynamics. It is not clear if SAC activation also leads to activation of MAPK pathways for them to contribute to the SAC robustness. To tie this loose end, the author could have checked if MAPK pathway is also activated under the conditions when SAC is activated. Unless this is shown, one must assume that the authors are attributing the effect they observe to the basal activity of MAPKs.

(iv) This is also a loose end:

The authors show that activation of stress pathways (by addition of KCL instance) causes phosphorylation-dependent Slp1Cdc20 downregulation (Figure 6) under SAC-activating conditions. Does activation of the stress pathway cause phosphorylation-dependent Slp1Cdc20 downregulation under non-SAC-activation conditions or does it occur only under SAC-activating conditions?

(v) Although the authors have gone to some length to identify S28, T31 (also S59) as phosphorylation sites for Cdk1, their functional significance in the context of MAPK involvement is not yet clear. Perhaps it is outside the scope of this study to dig deeper into this aspect more than the authors have.

(vi) In its current state, the Discussion section is quite disjointed. The first section "Involvement of MAPKs in cell cycle regulation" should be in the Introduction section (very briefly, if at all). It certainly does not belong to the Discussion section. In any case, the Discussion section should be more organized with better flow of arguments/interpretations.

---

## [Referee Report · Reviewer #2 (Public review)]

Summary:

This study by Sun et al. presents a role for the *S. pombe* MAP kinase Pmk1 in the activation of the Spindle Assembly Checkpoint (SAC) via controlling the protein levels of APC/C activator Cdc20 (Slp1 in S. pombe). The data presented in the manuscript is thorough and convincing. The authors have shown that Pmk1 binds and phosphorylates Slp1, promoting its ubiquitination and subsequent degradation. Since Cdc20 is an activator of APC/C, which promotes anaphase entry, constitutive Pmk1 activation leads to an increased percentage of metaphase-arrested cells. The authors have used genetic and environmental stress conditions to modulate MAP kinase signalling and demonstrate their effect on APC/C activation. This work provides evidence for the role of MAP kinases in cell cycle regulation in S. pombe and opens avenues for exploration of similar regulation in other eukaryotes.

Strengths:

The authors have done a very comprehensive experimental analysis to support their hypothesis. The data is well represented, and including a model in every figure summarizes the data well.

Weaknesses:

As mentioned in the comments, the manuscript does not establish that MAP kinase activity leads to genome stability when cells are subjected to genotoxic stressors. That would establish the importance of this pathway for checkpoint activation.

---

## [Author Response]

The following is the authors’ response to the original reviews.

**Public Reviews:**

**Reviewer #1 (Public Review):**
Summary:This manuscript addresses two main issues:(i) do MAPKs play an important role in SAC regulation in single-cell organism such as *S. pombe*?(ii) what is the nature of their involvement and what are their molecular targets?The authors have extensively used the cold-sensitive β-tubulin mutant to activate or inactivate SAC employing an arrest-release protocol. Localization of Cdc13 (cyclin B) to the SPBs is used as a readout for the SAC activation or inactivation. The roles of two major MAPK pathways i.e. stress-activated pathway (SAP) and cell integrity pathway (CIP), have been explored in this context (with CIP more extensively than SAP). sty1Δ or pmk1Δ mutants were used to inactivate the SAP or CIP pathways and wis1DD or pek1DD expression was utilized to constitutively activate these pathways, respectively. Lowering of Slp1Cdc20 abundance (by phosphorylation of Slp1-Thr 480) is revealed as the main function of MAPK to augment the robustness of the spindle assembly checkpoint.Strengths:The experiments are generally well-conducted, and the results support the interpretations in various sections. The experimental data clearly supports some of the key conclusions:(1) While inactivation of SAP and CIP compromises SAC-imposed arrest, their constitutive activation delays the release from the SAC-imposed arrest.(2) CIP signaling, but not SAP signaling, attenuates Slp1Cdc20 levels.(3) Pmk1 and Cdc20 physically interact and Pmk1-docking sequences in Slp1 (PDSS) are identified and confirmed by mutational/substitution experiments.(4) Thr480 (and also S76) is identified as the residue phosphorylated by Pmk1. S28 and T31 are identified as Cdk1 phosphorylation sites. These are confirmed by mutational and other related analyses.(5) Functional aspects of the phosphorylation sites have been elucidated to some extent: (a) Phosphorylation of Slp1-T480 by Pmk1 reduces its abundance thereby augmenting the SAC-induced arrest; (b) S28, T31 (also S59) are phosphorylated by Cdk1; (c) K472 and K479 residues are involved in ubiquitylation of Slp1.Weaknesses:(1) Cdc13 localization to SPBs has been used as a readout for SAC activation/inactivation throughout the manuscript. However, the only image showing such localization (Figure 1C) is of poor quality where the Cdc13 localization to SPBs is barely visible. This should be replaced by a better image.

We have replaced those pictures with a new set of representative images, which show clear presence or absence of SPB-localized Cdc13-GFP.

(2) The overlapping error bars in Cdc13-localization data in some figures (for instance Figure 3E and 4H) make the effect of various mutations on SAC activation/inactivation rather marginal. In some of these cases, Western-blotting data support the authors' conclusions better.

We agree that the overlapping error bars may look ambiguous in most figures showing time course curves, this is due to the fact that all these data from a group of strains have to be better presented in a single graph to more directly compare the potential effects. We have been fully aware of the drawback of these figure representations, that is why we always presented the data corresponding two major time points (0 and 50 min after release) from all time course analyses in an alternative way, namely using individual histograms to represent the data from each strain with means of repeats, absolute values, error bars and p values clearly labeled. In particular, the data from time point 0 min can provide important information on the SAC activation efficiency. Generally, we placed those data and graphs in corresponding supplemental figures, such as: Figure 1-figure supplement 1C, Figure 1-figure supplement 2D, Figure 3-figure supplement 3, Figure 4-figure supplement 6B, Figure 5-figure supplement 1, and Figure 6-figure supplement 2.

In addition, as you have noticed, almost all time course data were backed up by our Western blotting data.

(3) This specific point is not really a weakness but rather a loose end:One of the conclusions of this study is that MAPK (Pmk1) contributes to the robustness of SAC-induced arrest by lowering the abundance of Slp1Cdc20. The authors have used pmk1Δ or constitutively activating the MAPK pathways (Pek1DD) and documented their effect on SAC activation/inactivation dynamics. It is not clear if SAC activation also leads to activation of MAPK pathways for them to contribute to the SAC robustness. To tie this loose end, the author could have checked if the MAPK pathway is also activated under the conditions when SAC is activated. Unless this is shown, one must assume that the authors are attributing the effect they observe to the basal activity of MAPKs.

We agree with your concern. We have followed your suggestion and performed further experiments. Please see our more detailed response to your point #ii(a) in your “Recommendations for the authors”.

(4) This is also a loose end:The authors show that activation of stress pathways (by addition of KCl for instance) causes phosphorylation-dependent Slp1Cdc20 downregulation (Figure 6) under the SAC-activating condition. Does activation of the stress pathway cause phosphorylation-dependent Slp1Cdc20 downregulation under the non-SAC-activation condition or does it occur only under the SAC-activating condition?

We agree with your concern. We have followed your suggestion and performed further experiments. Please see our more detailed response to your point #ii(b) in your “Recommendations for the authors”.

(5) Although the authors have gone to some length to identify S28 and T31 (also S59) as phosphorylation sites for Cdk1, their functional significance in the context of MAPK involvement is not yet clear. Perhaps it is outside the scope of this study to dig deeper into this aspect more than the authors have.

Based on our data from Mass spectrometry analysis, mutational analysis, in vitro and in vivo kinase assays using phosphorylation site-specific antibodies, we confirmed that at least S28 and T31 are Cdk1 phosphorylation sites. From our time course analysis of these phosphorylation-deficient mutants, it seems the mechanisms of Slp1 activity or protein abundance regulated by Cdk1 or MAPK are quite different. How these two or even more kinases coordinate to control Slp1 activity during APC/C activation is one very interesting issue to be investigated, however, as you have realized, it is indeed beyond the scope of our current study.

(6) In its current state, the Discussion section is quite disjointed. The first section "Involvement of MAPKs in cell cycle regulation" should be in the Introduction section (very briefly, if at all). It certainly does not belong to the Discussion section. In any case, the Discussion section should be more organized with a better flow of arguments/interpretations.

We have re-organized our “Discussion” section. Please see our more detailed response to your point #iii in your “Recommendations for the authors”.

**Reviewer #2 (Public Review):**
Summary:This study by Sun et al. presents a role for the *S. pombe* MAP kinase Pmk1 in the activation of the Spindle Assembly Checkpoint (SAC) via controlling the protein levels of APC/C activator Cdc20 (Slp1 in S. pombe). The data presented in the manuscript is thorough and convincing. The authors have shown that Pmk1 binds and phosphorylates Slp1, promoting its ubiquitination and subsequent degradation. Since Cdc20 is an activator of APC/C, which promotes anaphase entry, constitutive Pmk1 activation leads to an increased percentage of metaphase-arrested cells. The authors have used genetic and environmental stress conditions to modulate MAP kinase signalling and demonstrate their effect on APC/C activation. This work provides evidence for the role of MAP kinases in cell cycle regulation in S. pombe and opens avenues for exploration of similar regulation in other eukaryotes.Strengths:The authors have done a very comprehensive experimental analysis to support their hypothesis. The data is well represented, and including a model in every figure summarizes the data well.Weaknesses:As mentioned in the comments, the manuscript does not establish that MAP kinase activity leads to genome stability when cells are subjected to genotoxic stressors. That would establish the importance of this pathway for checkpoint activation.We understand your concern. We have followed your suggestion and performed further experiments to examine whether the absence of Pmk1 causes chromosome segregation defects. Please see our more detailed response to your point #5 in your “Recommendations for the authors”.
**Recommendations for the authors:**

**Reviewing Editor**
Please go through the reviews and recommendations and revise the paper accordingly. I think nearly everything is very straightforward and all issues raised by the two expert referees are fully justified. I look forward to seeing an appropriately revised manuscript.
**Reviewer #1 (Recommendations For The Authors):**
(i) Cdc13 localization to SPBs has been used as a readout for SAC activation/inactivation throughout the manuscript. However, the only image showing such localization (Figure 1C) is of poor quality where the Cdc13 localization to SPBs is barely visible. This should be replaced by a better image.

We have replaced those pictures with a new set of representative images, which show clear presence or absence of SPB-localized Cdc13-GFP.

(ii) I reiterate the loose ends in this manuscript I have mentioned above. If the authors have already conducted these experiments, they should include the results in the manuscript to tighten the story further. (I am not suggesting that the authors must perform these experiments...if they have not).(a) One of conclusions of this study is that MAPK (Pmk1) contributes to the robustness of SAC-induced arrest by lowering the abundance of Slp1Cdc20. The authors have used pmk1Δ or constitutively activating the MAPK pathways (pek1DD) and documented their effect on SAC activation/inactivation dynamics. It is not clear if SAC activation also leads to activation of MAPK pathways for them to contribute to the SAC robustness. To tie this loose end, the author could have checked if the MAPK pathway is also activated under the conditions when SAC is activated. Unless this is shown, one must assume that the authors are attributing the effect they observe to the basal activity of MAPKs.

Actually, our data shown in Figure 6B demonstrated that SAC activation per se cannot trigger activation of MAPK pathway CIP, because we did not observe any elevated Pmk1 phosphorylation (i.e. Pmk1-P detected by anti-phospho p42/44 antibodies) in nda3-arrested cells (Please see “control” samples in Figure 6B).

To corroborate this observation, we further examined the Pmk1 phosphorylation/activation in Mad2-overexpressing cells, and could not detect elevated Pmk1 phosphorylation. This data again lends support to the notion that SAC activation per se cannot trigger activation of CIP signaling.

We have added our newly obtained result in Figure 6-figure supplement 1 in our revised manuscript.

(b) The authors show that activation of stress pathways (by addition of KCL instance) causes phosphorylation-dependent Slp1Cdc20 downregulation (Figure 6) under the SAC-activating conditions. Does activation of the stress pathway cause phosphorylation-dependent Slp1Cdc20 downregulation under the non-SAC-activation conditions or does it occur only under the SAC-activating condition?

As you suggested, we have constructed cdc25-22 background strains with pmk1+ deleted or expressing Padh11-pek1DD to remove or constitutively activate CIP signaling, respectively. By immunoblotting, we followed the Slp1Cdc20 levels when cells went through mitosis after being released at 25 °C from G2/M-arrest at high temperature. We found that Slp1Cdc20 levels in pek1DD cells were only marginally reduced compared to wild-type cells, whereas we failed to observe any elevated Slp1Cdc20 levels in pmk1Δ cells. These results suggested that CIP signaling only plays a negligible role in influencing Slp1Cdc20 levels under the non-SAC-activation conditions.

We have presented our newly obtained result in Figure 2-figure supplement 1 in our revised manuscript.

(iii) The Discussion section is quite disjointed. The first section "Involvement of MAPKs in cell cycle regulation" should be in the Introduction section (very briefly, if at all). It certainly does not belong to the Discussion section. In any case, the Discussion section should be more organized with a better flow of arguments/interpretations.

Thank you for suggestion on the organization and flow for “Discussion”. We have reorganized our “Discussion” sections and moved the previous “Involvement of MAPKs in cell cycle regulation” to the section “Introduction” and rewrote the corresponding paragraph.

(iv) A minor point in this context:In the cold-sensitive β-tubulin mutant, growth at 18C causes loss of kinetochore-microtubule attachments as well as the intra-kinetochore tension. Both perturbations individually can lead to the activation of SAC. This study does not distinguish whether MAPK involvement in SAC dynamics is relevant to one perturbation or another or both. It would be pertinent to briefly mention this point in the Discussion section.

As you suggested, we have added two sentences to briefly mention this point in our “Discussion” section.

**Reviewer #2 (Recommendations For The Authors):**
This study by Sun et al. presents a role for the *S. pombe* MAP kinase Pmk1 in the activation of the Spindle Assembly Checkpoint (SAC) via controlling the protein levels of APC/C activator Cdc20 (Slp1 in S. pombe). The data presented in the manuscript is thorough and convincing. The authors have shown that Pmk1 binds and phosphorylates Slp1, promoting its ubiquitination and subsequent degradation. Since Cdc20 is an activator of APC/C, which promotes anaphase entry, constitutive Pmk1 activation leads to an increased percentage of metaphase-arrested cells. The authors have used genetic and environmental stress conditions to modulate MAP kinase signalling and demonstrate their effect on APC/C activation. This work provides evidence for the role of MAP kinases in cell cycle regulation in S. pombe and opens avenues for exploration of similar regulation in other eukaryotes.Although the data largely supports the conclusions, a major addition will be testing whether cells accumulate chromosomal or inheritance defects when MAPK Pmk1 is absent. It will be interesting to know that this mechanism of SAC activation contributes to genome integrity.Some additions that can improve the manuscript are mentioned below:(1) In Figure 1, the authors should also test the effect of constitutive activation of Spk1 to rule out the involvement of the PSP pathway.

To meet your curiosity and requirement, we have constructed yeast strains expressing constitutively active *byr1DD* alleles carrying S214D and T218D point mutations under the control of the *adh21* or *adh11* promoters (*Padh21* or *Padh11* in short), i.e. *Padh21-6HA-byr1DD* and *Padh11-6HA-byr1DD*, respectively. We examined the expression of these *byr1DD* alleles by Western blotting, and tested the TBZ sensitivity of these alleles and also checked whether they affect the efficiency of SAC activation or inactivation. Our results showed that constitutive activation of Spk1 by overexpressing Byr1DD does not cause yeast cells to be TBZ-sensitive or affect the efficiency of SAC activation or inactivation.

We have added these new data in Figure 1-figure supplement 2 in our revised manuscript.

(2) The number of analyzed cells (n) should be mentioned in the figure legends in Figure 1D, and all other figure panels should represent similar data in the consequent figures.

We have added the information on sample size for all experiments involving time course analyses.

(3) The authors should also use another arresting mechanism (e.g. nocodazole treatment) and corroborate the result in Figure 1C to rule out any effects due to the mutant.

Figure 1C in our manuscript actually shows our experimental design and not the result. We guess here you asked for alternative strategy to arrest cells at metaphase and confirm our results shown in Figure 1D.

We need to mention that, as a commonly used inhibitor of microtubule polymerization, Nocodazole is very effective in mammalian and human cells and also in budding yeast cells, but not effective at all in wild-type fission yeast cells. It has been found that Nocodazole is only active in fission yeast α- or β-tubulin mutants (please see Umesono, K., *et al*., J Mol Biol. 168 (2): 271-284 (1983); PMID: 6887245; DOI: 10.1016/s0022-2836(83)80018-7.) or multidrug resistance (MDR) transporter mutants (please see Kawashima, SA, *et al.*, Chemistry & Biology 19, 893–901 (2012); PMID: 22840777; doi: 10.1016/j.chembiol.2012.06.008.). Therefore, this feature of Nocodazole has limited and restricted its routine use as a metaphase arrest or spindle checkpoint activation drug in fission yeast.

Instead, in order to achieve the spindle checkpoint activation and metaphase arrest, we took advantage of a metaphase-arresting mechanism involving Mad2 overexpression, which has been described and used previously (Please see He, X., *et al*., Proc Natl Acad Sci USA. 94 (15): 7965-70 (1997); PMID: 9223296; DOI: 10.1073/pnas.94.15.7965, and May, K.M., *et al*., Current Biology, 27(8):1221-1228 (2017); PMID: 28366744; DOI: 10.1016/j.cub.2017.03.013). With this strategy, we could analyze the metaphase-arresting and SAC-activation efficiency by counting cells with short spindles as judged by GFP-Atb2 signals. We compared the frequencies of cells with short spindles in wild-type, *pmk1Δ*, *sty1-T97A*, and *spk1Δ* backgrounds after Mad2 has been induced to overexpress for 18 hours, and found that SAC-activating efficiency was compromised in *pmk1Δ* and *sty1-T97A* cells, but not in *spk1Δ* cells. This data indeed corroborated our result shown in Figure 1D and ruled out possible effects due to the *nda3-KM311* mutant.

We have added this new data in Figure 1-figure supplement 3 in our revised manuscript.

(4) It would also be helpful to assess SAC or APC/C activation with another cellular readout in addition to Cdc13-GFP accumulation on SPBs, at least for initial experiments.

Actually, Cdc13-GFP accumulation on SPBs has been routinely used as a reliable cellular readout for SAC or APC/C activation in the field. It was first developed and used by Kevin Hardwick lab in their paper (Vanoosthuyse V and Hardwick KG. Curr Biol. 2009, 19(14):1176-81. PMID: 19592249; doi: 10.1016/j.cub.2009.05.060.). This method was also used in a paper by Meadows JC, et al. (2011) (Dev Cell. 20(6):739-50. PMID: 21664573; doi: 10.1016/j.devcel.2011.05.008.).

In our previous study, we also employed a different strategy to assess SAC inactivation or APC/C activation, in which degradation of nuclear Cut2-GFP was used as a cellular readout (Please see S4 Fig in Bai S, *et al*., PLoS Genet 18(9): e1010397 (2022); PMID: 36108046; DOI: 10.1371/journal.pgen.1010397.). Cut2 is the securin homologue in *S. pombe* and therefore also a target of APC/C at anaphase. Our data in the above paper confirmed that the degradation of both nuclear Cut2-GFP and SPB-localized Cdc13-GFP shows similar dynamics when cells are released from metaphase-arrest.

As we described in our response to your comment #3, we employed short spindles visualized by GFP-Atb2 signals as an alternative readout for metaphase-arrest and SAC-activation in cells overexpressing Mad2. We confirmed that SAC-activation efficiency was compromised in *pmk1Δ* and *sty1-T97A* cells, but not in *spk1Δ* cells.

(5) The authors have shown a role for Pmk1 in controlling the activation of APC/C and, hence, cell cycle progression through metaphase to anaphase. One crucial experiment would be to check if pmk1Δ cells show an accumulation of chromosomal aberrations or unequal distribution when subjected to genotoxic stressors. That would implicate a direct importance on Pmk1's role in cell cycle arrest and genome maintenance.

As you suggested, we have constructed *cdc25-22 GFP-atb2+* strains with *pmk1+* present or deleted, and treated cells with 0.6 M KCl or 2 μg/mL caspofungin to activate MAPKs and checked whether the absence of *pmk1* could cause defective chromosome segregation in anaphase cells. Indeed, we found that stressed *pmk1Δ* cells displayed greatly increased frequency of lagging chromosomes and chromosome mis-segregation at mitotic anaphase compared to similarly treated wild-type cells and also untreated *pmk1Δ* cells. This new data implicated a direct role for Pmk1 in cell cycle arrest and genome maintenance, especially when cells are exposed to adverse environment.

We have presented this new data as Figure 7 in our revised manuscript.

Typos:(1) In line 406, "docking" is misspelled as "docing".

Thank you for pointing this out. We have corrected this mistake.

(2) In Figure 6, panel "F" is not marked in the figure.

We mistakenly mentioned and labeled “F” in Figure 6 legend. In our revised manuscript, we have added new results of protein levels of Pmk1 phosphorylation- and ubiquitylation-deficient Slp1Cdc20 mutants upon SAC activation detected by Western blotting in Figure 6-figure supplement 3.

(3) In Figure S1, panel "D" is not marked.

We apologize for our previous wording in our former Figure S1 legend, which was misleading. Actually, there was no panel “D” in Figure S1 (now Figure 1-figure supplement 1). We have rewritten the legend to avoid ambiguity.